# Distributed genetic architecture across the hippocampal formation implies common neuropathology across brain disorders

Shahram Bahrami [1✉], Kaja Nordengen [1,2], Alexey A. Shadrin[1,3], Oleksandr Frei[1,4], Dennis van der Meer[1,5], Anders M. Dale [6,7,8], Lars T. Westlye [1,3,9], Ole A. Andreassen [1,3] & Tobias Kaufmann [1,10✉]

Despite its major role in complex human functions across the lifespan, most notably navigation, learning and memory, much of the genetic architecture of the hippocampal formation is currently unexplored. Here, through multivariate genome-wide association analysis in volumetric data from 35,411 white British individuals, we reveal 177 unique genetic loci with distributed associations across the hippocampal formation. We identify genetic overlap with eight brain disorders with typical onset at different stages of life, where common genes suggest partly age- and disorder-independent mechanisms underlying hippocampal pathology.

[1] Norwegian Centre for Mental Disorders Research, Division of Mental Health and Addiction, Oslo University Hospital & Institute of Clinical Medicine, University of Oslo, Oslo, Norway. [2] Department of Neurology, Oslo University Hospital, Oslo, Norway. [3] KG Jebsen Centre for Neurodevelopmental Disorders, University of Oslo, Oslo, Norway. [4] Department of Informatics, University of Oslo, Oslo, Norway. [5] School of Mental Health and Neuroscience, Faculty of Health, Medicine and Life Sciences, Maastricht University, Maastricht, The Netherlands. [6] Department of Radiology, School of Medicine, University of California, San Diego, CA, USA. [7] Department of Neurosciences, University of California San Diego, La Jolla, CA, USA. [8] Center for Multimodal Imaging and Genetics, University of California at San Diego, La Jolla, CA, USA. [9] Department of Psychology, University of Oslo, Oslo, Norway. [10] Department of Psychiatry and Psychotherapy, Tübingen Center for Mental Health, University of Tübingen, Tübingen, Germany. ✉email: shahram.bahrami@medisin.uio.no; tobias.kaufmann@med.uni-tuebingen.de

The hippocampal formation plays critical roles in episodic memory[1,2], navigation[3,4], and emotions[5]. Consequently, impaired or lesion-induced loss of hippocampal functioning has tremendous and diverse impact on emotions and cognitive functions[6,7]. Thus, the hippocampus has been extensively studied across a wide variety of diseases and traits, including trajectories of early development and aging.

The degree of hippocampal maturation and change throughout the lifespan may capture information relevant to the study of psychiatric and neurological disorders, where the age at which individual neurophysiological trajectories usually diverge from the norm may reflect key characteristics of the underlying pathophysiology[8]. Indeed, the hippocampal formation is known to be involved in disorders with typical onset early in life[9], such as autism spectrum disorders (ASD), attention-deficit hyperactivity disorder (ADHD), schizophrenia (SCZ) and bipolar disorder (BIP), through its roles in perception, memory processes, modulation of executive function, emotion regulation, among others[10–12]. Furthermore, through its involvement in stress response, the hippocampal formation is potentially involved in migraine (MIG)[13] and tests of recollection memory indicate hippocampal dysfunction in major depression (MD)[14], both of which are disorders that can appear at any stage from adolescence to old age. Finally, the hippocampal formation is implied in diseases that primarily emerge during senescence such as Parkinson's disease (PD) and Alzheimer's disease (AD). Emerging data suggests a complex hippocampal crosstalk among the dopaminergic and other transmitter systems in PD, where the hippocampal formation is involved in adaptive memory and motivated behaviour[15]. Loss of hippocampal functions like navigation and episodic memory are core markers of AD and hippocampal atrophy is an established finding[16,17]. Together, these studies highlight the role of the hippocampal formation in a range of psychiatric and neurological disorders across the lifespan.

The past decades have brought significant progress towards a characterization of the genetic architecture of the hippocampal formation, from experimental manual mapping in mice[18] to brain imaging-based genome-wide-association studies (GWAS) in humans[19–22], initially based on total hippocampus volume reporting one[20] and six[19] loci, and further increasing to 15 when studying individual hippocampal subfields[21]. Given the broad functional portfolio of the hippocampal formation, however, it is clear that much of the genetic architecture remains to be explored, calling for further studies and novel analytical approaches[23].

Recent work revealed a distributed genetic architecture of human brain anatomy[24,25] and function[26] and suggested that capitalizing on this distributed nature in a multivariate GWAS approach can significantly improve the discovery beyond standard GWAS approaches[24]. Advancements into deriving subdivisions of the hippocampus through adaptive segmentation has allowed for a fine-grained assessment across multiple subregions[27]. We hypothesized that the genetic architecture within the hippocampal formation is distributed across its subregions and thus aimed to gain novel insights into the genetics of the hippocampal formation by deploying such multivariate GWAS approach to the 19 subregion volumes that can currently be segmented with MRI[27]. Further, given hippocampal involvement in many severe and highly prevalent brain disorders across the lifespan, we targeted common neurological and psychiatric disorders ranging from developmental disorders to neurodegenerative disorders, aiming to reveal gene variants potentially involving the hippocampus at different stages in life.

We accessed raw T1-weighted MRI data from 35,411 genotyped white British individuals (age range: 45–82 years, mean: 64.4 years, s.d.: 7.5 years, 51.7% females) from the UK Biobank[28]

(permission no. 27412) and segmented the hippocampal formation into 19 subregions in addition to total hippocampus volume (sum of all subfields) using FreeSurfer 7.1[27] (Fig. 1a). For each of these, we calculated the average volume between the left and right hemisphere and residualized for age, age squared, sex, scanning site, a proxy of image quality, intracranial volume and the first 20 genetic principal components. The resulting residuals were used in genetic analyses, feeding the 19 subregions alongside whole hippocampus volume into the Multivariate Omnibus Statistical Test (MOSTest)[24], which implements permutation testing to identify genetic effects across multiple phenotypes, yielding a multivariate GWAS summary statistic across all 20 features.

## Results

**Multivariate approach identifies 177 loci associated with the hippocampal formation.** In line with our hypothesis, we found strong support of a distributed genetic architecture in the hippocampal formation. Multivariate GWAS revealed 177 unique genetic loci with distributed associations across the hippocampal formation. The upper part of Fig. 1b depicts the corresponding multivariate statistics, highlighting the polygenic architecture of the hippocampal formation. For each of the 177 loci, the lower part of Fig. 1b depicts statistics from univariate GWASs of individual hippocampal subregions. The elevated univariate statistics for multiple hippocampal subregions in some of the same loci supports a distributed genetic architecture across the hippocampal formation, which is also supported by genetic correlation analysis of the univariate GWASs of the individual subregions (Supplementary Fig. 1, Supplementary Data 1). Whereas the strongest hits among the 177 discovered loci are also implied in univariate analysis, a large share of the 177 loci showed elevated yet not genome-wide significant effects at univariate level. By capitalizing on these distributed effects across subregions, the multivariate approach boosted discovery. Supplementary Data 2 provides additional details on the 177 discovered loci, most of which were not identified in previous hippocampus GWAS. Supplementary Fig. 2 depicts corresponding quantile–quantile (Q–Q) plots, including one from permutation testing that confirms validity of the multivariate test statistic. A multivariate replication attempt in an independent sample of 5262 individuals with non-white ethnicity supports robustness of the findings, yielding same effect direction for 98% of the lead SNPs (Supplementary Fig. 3).

**Functional mapping and annotation identifies 87 genes robustly associated with the hippocampal formation.** We functionally annotated all candidate SNPs ($n = 25704$) that were in linkage disequilibrium ($r^2 \geq 0.6$) with one of the independent significant SNPs using functional mapping and annotation of GWAS (FUMA)[29]. About 90% of the SNPs had a minimum chromatin state of 1–7, thus suggesting they were in open chromatin regions[30,31], and 6.1% were in regulomeDB category 1 or 2, suggesting potential regulatory function[30] (Supplementary Fig. 4A, B). A majority of these SNPs were intronic (49.9%) or intergenic (30%) and 1.0% were exonic (Supplementary Fig. 4C).

We mapped the 177 loci implied for the hippocampal formation to 963 genes based on four different mapping strategies (positional, expression quantitative trait loci (eQTL), chromatin interaction mapping and MAGMA analysis). Out of these, 87 genes were identified by all four mapping strategies (Fig. 2), supporting robustness of these findings. Supplementary Data 3–6 and Supplementary Fig. 5 provide additional details. Four genes were mapped from the locus with strongest GWAS effect: *TESC*, important in cell proliferation and differentiation[20], *FBXW8*, important for ubiquitination and protease-mediated degradation[32]

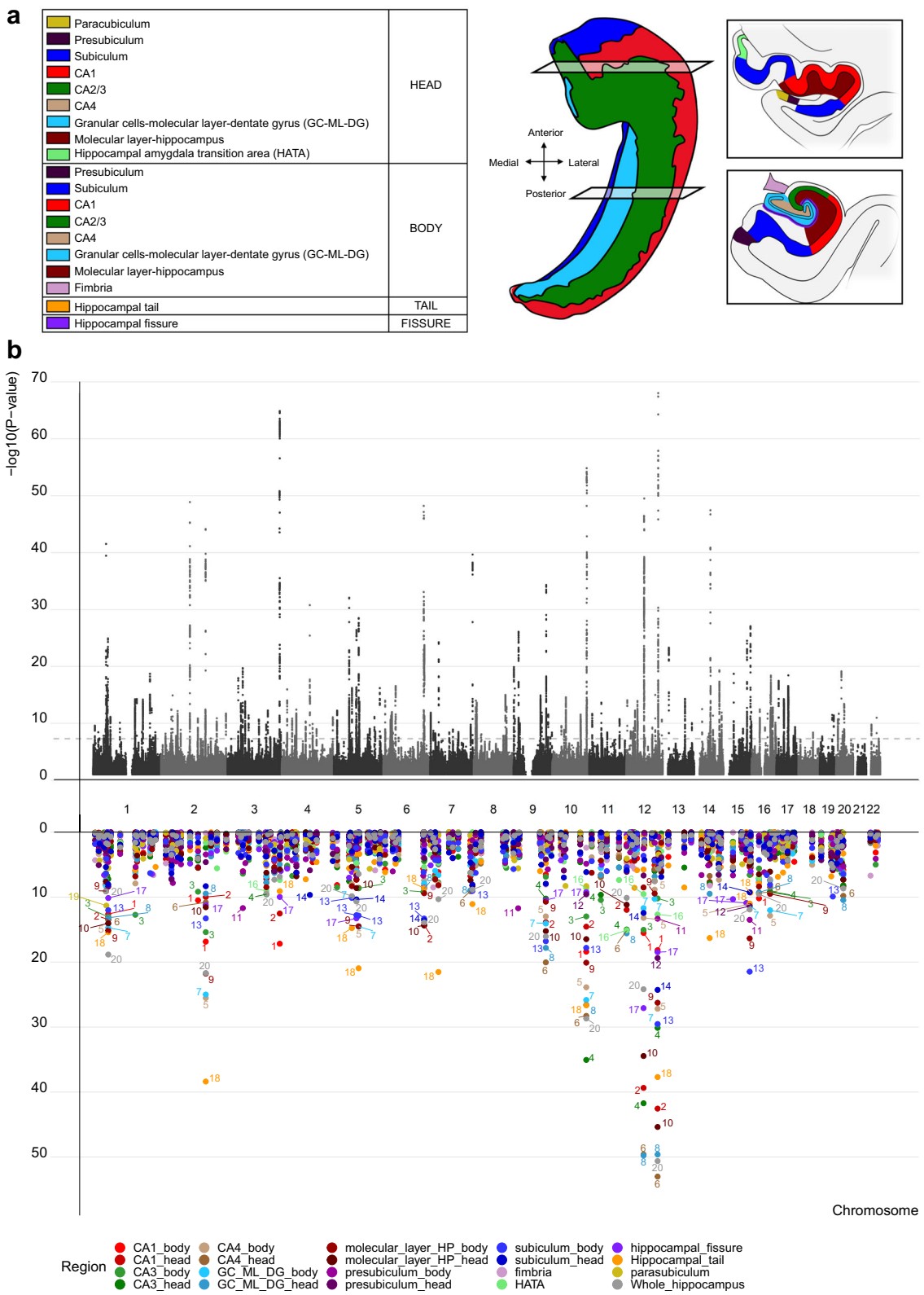

with a suggested role in clearing protein aggregates like hyperphosphorylated tau[33] in addition to synapse formation[34], *C12orf49*, an essential regulator of fatty acid metabolism[35], and finally *RNFT2* (also known as *TMEM118*), important for immune regulation[36]. The second strongest GWAS hit, mapped to *LEMD3* (also known as *MAN1*) is also relevant for immune regulation[37]. *FAM53B*, mapped from the third, *POU3F3*, mapped from the fourth, and

*VCAN*, mapped from the fifth strongest locus, all have important roles in neurodevelopment[38–40].

Genome-wide gene-based association studies (GWGAS; two-sided $P < 2.7 \times 10^{-6}$) through MAGMA identified 303 unique genes across the hippocampus (Supplementary Data 5). Many of the significant gene sets reflected processes related to early development (Supplementary Data 7), such as neurogenesis

**Fig. 1 The multivariate framework discovered 177 independent loci significantly associated with the hippocampal formation. a** Schematic illustration of the hippocampus regions. The hippocampal formation comprises the histologically distinguishable subfields of the hippocampus proper as well as the dentate gyrus with its own subfields, and the neocortical subiculum, presubiculum and parasubiculum. In all but the latter, the hippocampal formation is also divided into an anterior (head) and a posterior part (body). **b** The upper part illustrates the $-\log_{10}(P)$ statistic from the multivariate GWAS across the entire formation, with 177 significant loci. The lower part depicts for each of the 177 unique loci the corresponding $-\log_{10}(P)$ statistics from univariate GWASs of single subregions (one colour per subregion, $p$-values are two-tailed), supporting a distributed genetic architecture across the hippocampal formation. The strongest effects are labelled with numbers that reflect regions according to the order of regions in the legend (1 = CA1_body, 2 = CA1_head, 3 = CA3_body, 4 = CA3_head, 5 = CA4_body, 6 = CA4_head, 7 = GC_ML_DG_DG_body, 8 = GC_ML_DG_head, 9 = molecular_layer_HP_body, 10 = molecular_layer_HP_head, 11 = presubiculum_body, 12 = presubiculum_head, 13 = subiculum_body, 14 = subiculum_head, 15 = fimbria, 16 = HATA, 17 = hippocampal fissure, 18 = Hippocampal_tail, 19 = parasubiculum, 20 = Whole_hippocampus).

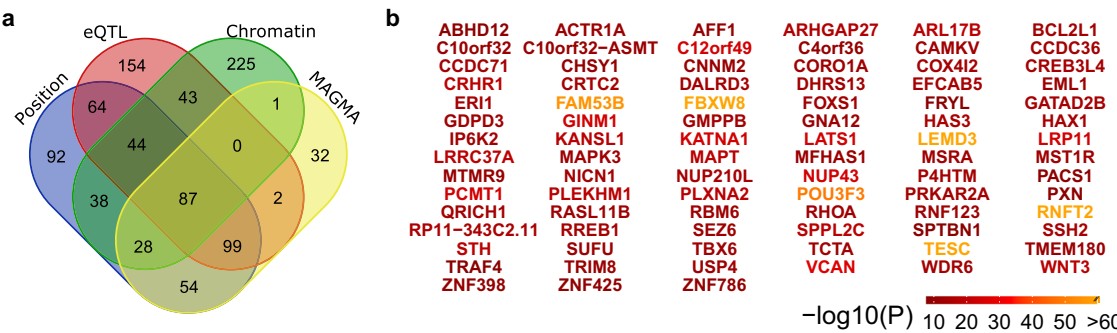

**Fig. 2 Gene mapping of the 177 loci associated with the hippocampal formation implied 87 genes by all mapping strategies. a** Venn diagram showing number of genes mapped by the four different strategies. **b** The 87 genes implied by all four strategies with colour-coded $-\log_{10}(P)$ statistics from the multivariate GWAS.

$(P_{\text{Bonf}} = 9.2 \times 10^{-8})$, regulation of anatomical structure morphogenesis $(P_{\text{Bonf}} = 1.3 \times 10^{-8})$ and neuronal differentiation $(P_{\text{Bonf}} = 1.6 \times 10^{-6})$, potentially indicating that individual differences in hippocampal volumes later in life may be largely determined early in development. Also, when focusing on the 87 genes implicated from all four mapping strategies (see above), the gene-sets reflect processes related to early development like regulation of cell morphogenesis $(P = 2.7 \times 10^{-7}$; Supplementary Data 8). Overrepresented pathways for the 87 common mapped genes included pathways representing processes prominent during the lifetime, from brain development (e.g. axon guidance, neuronal migration, angiogenesis) to plasticity (long-term depression) and finally damage mechanisms (protein repair, spinal cord injury, regulation of bad phosphorylation and neurodegeneration). (Supplementary Data 9). An analysis of cell types[41] for the 87 genes implicated strongest expression in foetal astrocytes, followed by mature astrocytes and neurons (Supplementary Fig. 6).

**Genetic overlap between the hippocampal formation and common brain disorders.** We studied the genetic overlap between hippocampal formation and eight disorders: ASD, ADHD, SCZ, BIP, MIG, MD, PD, and AD. By choosing disorders ranging from developmental disorders to neurodegenerative disorders, we make sure to cover a wide range of biological processes potentially involving the hippocampal formation at different stages in life. The commonly used approach, genetic correlations of the disorders with individual hippocampus subregions, did not show significant associations after Bonferroni correction for multiple comparisons (Supplementary Figs. 7 and 8 and Supplementary Data 10). However, conditional Q–Q plots[42] conditioning the multivariate statistic of hippocampal formation on the disorders and vice versa showed a clear pattern of pleiotropic enrichment in both directions (Supplementary Fig. 9). Conjunctional FDR analysis[42,43] allowed us to test for shared loci between the hippocampus and each of the disorders. Strikingly, we identified 8 loci significantly (conjFDR < 0.05) overlapping with

ADHD, 4 loci with ASD, 77 with BIP, 161 with SCZ, 41 with MD, 80 with MIG, 19 with AD and 10 loci significantly overlapping with PD (Fig. 3a).

Supplementary Data 11–18 provide a full list of loci overlapping between hippocampal formation and the disorders. We mapped each of these loci to genes using positional, eQTL and chromatin interaction mapping (Supplementary Data 19) and checked for genes that were implicated for multiple disorders. By far strongest overlap was found between SCZ and BIP, where 106 of the genes overlapping between hippocampal formation and SCZ were also found to overlap between hippocampal formation and BIP (Fig. 3b). While this overlap may be expected given the relatedness of the disorders, it is particularly noteworthy that we found large overlap between other combinations of disorders as well, some of which pertain very different onset times across the lifespan such as ASD and AD (14 genes), ADHD and PD (11 genes), or ASD and PD (14 genes). Many genes were implied for more than two disorders, and Fig. 3c depicts the subset implicated as overlapping with hippocampal formation for at least four disorders. Again, it is particular worth noting the co-occurance for distinct disorders in different phases of life. For example, the most frequently mapped gene was the *AMT* gene involved in glycinergic neurotransmission, found to overlap between hippocampal formation and ADHD, SCZ, MD, MIG and PD, respectively (Supplementary Data 19). Other examples are the tau protein associated genes *MAPT* and *STH*, found for ASD, SCZ, AD and PD, or the *GPX1* gene, known to protect cells from oxidative stress and here found for ADHD, SCZ, MD and PD. This may illustrate genetic mechanisms independent of life phases and may suggest that some of the pleiotropy between brain disorders might be explained by shared mechanisms in hippocampal pathology.

**Discussion**

Taken together, our multivariate GWAS of the volumes of the hippocampal formation revealed a plethora of genomic loci not

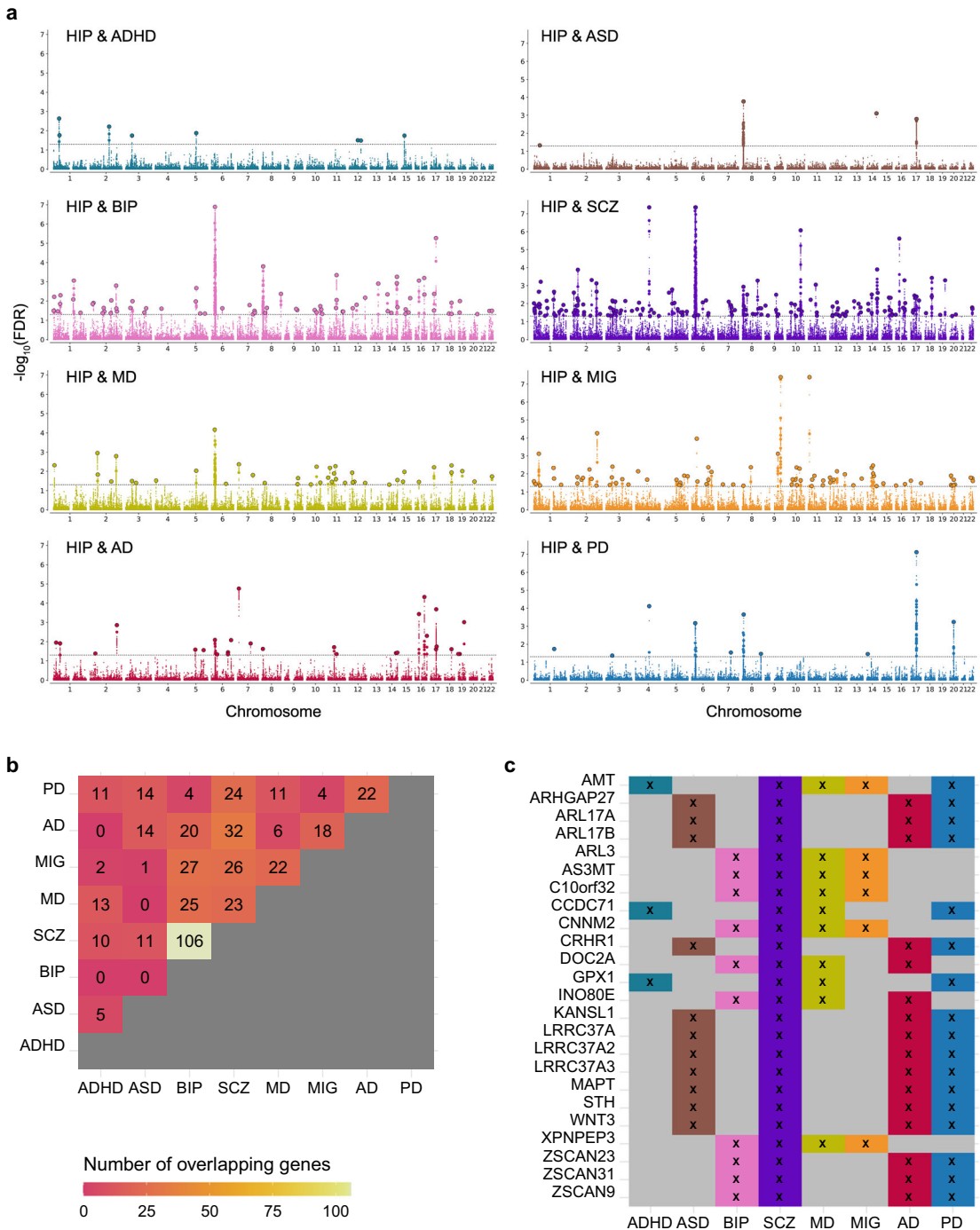

**Fig. 3 Genetic overlap between hippocampal formation and eight disorders with different onset times across the lifespan. a** For each disorder, a conjunctional FDR Manhattan plot is shown, illustrating the –log10 transformed conjunctional FDR values for each SNP on the *y*-axis and chromosomal positions along the *x*-axis. The dotted horizontal line represents the threshold for significant shared associations (conjFDR < 0.05). Independent lead SNPs are encircled in black. **b** Various genes mapped from the conjunctional FDR analysis were implied to overlap between hippocampal formation and multiple disorders. The figure illustrates the total number of genes implied in each combination of disorders. For example, 106 of the genes overlapping between hippocampal formation and SCZ were also found to overlap between hippocampal formation and BIP. **c** Panel **c** complements panel **b** with a list of genes that were implied for more than 3 disorders. One gene was mapped for five disorders (*AMT*) and 23 genes were mapped for four disorders. ASD autism spectrum disorder, ADHD attention deficit hyperactivity disorder, SCZ schizophrenia, BIP bipolar disorder, MIG migraine, MD, major depression, PD Parkinson's disease, AD Alzheimer's disease and HIP hippocampal formation.

identified in previous work, implicating a distributed nature of effects on hippocampus. The mapped genes have roles in neurobiological processes across the lifespan. Importantly, the profound overlap between hippocampal formation and common brain disorders and the identification of some of the same genes

implied for disorders with onset in different phases of life suggests age-independent neuropathology and may pinpoint potential disease-independent drug targets.

The distributed genetic architecture across the hippocampal formation, here revealed through 177 hippocampus-associated

loci and 87 genes mapped consistently by four mapping strategies, pointed at pathways with involvement across the lifespan, starting with embryogenic brain development, like axon guidance and neuronal migration, then involving neuronal plasticity processes, and finally pathways of neurodegeneration. Our findings align with and expand upon earlier reports from univariate analyses of the hippocampus[19–22], pointing at neurogenesis-related pathways[22] and linking hippocampal volumes to common brain disorders[19,21].

Our results suggest that several genes have a role in hippocampal pathology across multiple brain disorders, with onset times ranging across the lifespan. These findings not only support the notion of pleiotropy across a spectrum of neurological and psychiatric disorders but may also pinpoint to both, age- and disorder-independent drug targets. For example, our discovery of the *AMT* gene overlapping between hippocampal formation and ADHD, SCZ, MD, MIG and PD, respectively, may implicate an age-independent role of glycine in hippocampal pathology. The *AMT* gene codes for aminomethyltransferase (T-protein), an enzyme crucial for the glycine decarboxylase complex (GCS) in mitochondria. Glycine is a primary inhibitory neurotransmitter in the spinal cord and brainstem, but increasing evidence shows important glycine involvement also in the hippocampal formation[44]. Glycine exerts a tonic inhibitory role through extrasynaptic glycine receptor chloride channels[45], in addition to modulation of NMDA receptors in the hippocampal formation as evident from rodent studies[46,47]. Rodent studies also suggest that through regulation of both glycine and serine synthesis and cleavage, aminomethyltransferase as part of the glycine decarboxylase complex, may provide a homoeostatic regulation of hippocampal function and plasticity by simultaneous activation of excitatory NMDA receptors and inhibitory glycine receptors[48,49]. The glycine site on the NMDA receptors is currently under investigation as a promising drug target for several of the disorders that we here associated with the *AMT* gene, including ADHD[50,51], PD[52], SCZ[53] and MD[54], either by direct glycine supplementation, other substances working on the same receptor site or by increasing endogenous glycine by inhibiting the glycine transporter. Another example for a potential age- and disorder-independent drug target is the microtubule-associated protein tau (*MAPT*) gene, which was here implicated for ASD, SCZ, AD and PD. Indeed, tau has for long been a marker of AD and PD[55] yet has recently also gained focus for ASD[56], with animal models suggesting that tau reduction may prevent behavioural signs of this neurodevelopmental disorder[57]. Taken together, our results therefore add support for disorder-independent gene targets for hippocampal pathology across the lifespan, including *AMT* and *MAPT*, among others, and illustrate how the multivariate GWAS approach can reveal overlapping biochemical mechanisms underlying different disorders and traits.

Some aspects are relevant for interpreting the results of this study. First, there is currently no optimal method for gene mapping. We here chose to apply four different mapping strategies and highlighted genes that were identified in all four strategies. While this cannot fully overcome all limitations of current gene mapping approaches, we consider an identification of a given gene by four strategies as an indicator of robustness. Second, we identified genes associated with the hippocampal formation and several of the brain disorders across the lifespan, which might point to shared molecular pathways. However, it has to be emphasized that they could also have different roles in different pathologies. This study lays the foundation for future detailed studies of potential common pathways and drug targets. Third, generalizability of our findings beyond the study population remains to be investigated. We here performed our main analysis in data from 35,411 white British individuals and replicated our findings in data from 5262 individuals with non-white ethnicity, however, both samples were drawn from the same UK Biobank study population. Forth, it should be noted that the here used GWAS for major depression partly included samples that were not necessarily clinically diagnosed with major depressive disorder but reported symptoms of the disorder. This may factor into the specificity of this GWAS. Finally, future research targeting lateralization effects may yield additional insight into hemispheric similarities and differences in distributed genetic effects across regions of the hippocampal formation.

In conclusion, our results suggest a polygenic architecture of the hippocampal formation, distributed across its subregions. The genetic overlap with various brain disorders with typical onset at different stages of life implicated genes that may be relevant targets for future studies into the mechanisms underlying hippocampal functioning and pathology across the lifespan. With several of the findings fitting currently studied treatment targets (e.g. the glycine site on the NMDA receptor), our results also confirm the utility of the approach and suggest that capitalizing on the distributed nature of genetic effects on the brain will be instrumental in our future endeavours to further understand mechanisms underlying the brain and its disorders.

## Methods

**Sample and pre-processing of imaging and genetic data.** The UK Biobank was approved by the National Health Service National Research Ethics Service (ref. 11/NW/0382). We accessed raw T1-weighted magnetic resonance brain imaging data from 35,411 genotyped white British from the UK Biobank[28] (age range: 45–82 years, mean: 64.4 years, s.d.: 7.5 years, 51.7% females) for the main analysis, and of 5262 individuals with non-white ethnicity (age range: 45–81, mean: 62.9, s.d.: 7.6 years, 53.6% females) for the replication in independent data.

We processed T1-weighted images using the standard recon-all pipeline in Freesurfer 5.3[58], and subsequently segmented the hippocampal formation using Freesurfer 7.1[27]. The segmentation method has previously been validated in three independent data sets, attributing robust performance and highly replicable results[27]. For genetic analyses, we followed the standard quality control procedures to the UK Biobank v3 imputed genetic data and removed SNPs with an imputation quality score < 0.5, a minor allele frequency < 0.005, missing in more than 10% of individuals, and failing the Hardy–Weinberg equilibrium tests at a $P < 1e-9$.

**Multivariate genome-wide association analysis and gene mapping.** For each of the 19 regions of the hippocampal formation—parasubiculum, presubiculum head and body, subiculum head and body, CA1/CA3/CA4 head and body, GC-ML-DG head and body, molecular layer head and body, HATA, fimbria, hippocampal tail, hippocampal fissure—as well as for total hippocampus volume we calculated the average volume between the left and right hemisphere and subsequently residualized the volumes for age, age squared, sex, scanning site, Euler number as a proxy of image quality[59], intracranial volume and the first 20 genetic principal components. The resulting residuals for the 20 regions were jointly fed into the multivariate omnibus statistical test (MOSTest)[24] analysis. MOSTest implements permutation testing to identify genetic effects across multiple phenotypes, yielding a multivariate GWAS summary statistic across all 20 features. For mathematical details of the implementation, see van der Meer et al. (2020)[24], for details on the software implementation see github.com/precimed/mostest. MOSTest has been extensively validated in the original methods paper, including simulations and comparisons with other methods that have confirmed its solid performance in discovery and an order of magnitude shorter runtime compared to other tools[24]. For comparison to standard univariate approaches, we also performed univariate GWAS (extracted from the univariate stream of MOSTest[24]). Supplemental genetic correlation analyses were performed using LD-score regression[60,61]. Heritability was estimated using genome-wide complex trait analysis (GCTA)[62], and for comparison using LD-score regression[60,61].

To identify genetic loci we uploaded this summary statistic to the FUMA platform v1.3.7[29]. Using the 1000GPhase3 EUR as reference panel, we identified independent significant SNPs at the statistical significance threshold $P < 5e-8$. All SNPs at $r^2 < 0.6$ with each other were considered as independent significant SNPs and a fraction of the independent significant SNPs in approximate linkage equilibrium with each other at $r^2 < 0.1$ were considered as lead SNPs. FUMA annotates associated SNPs based on functional categories, Combined Annotation Dependent Depletion (CADD) scores which predicts the deleteriousness of SNPs on protein structure/function[63], RegulomeDB scores which predicts regulatory functions[30]; and chromatin states that shows the transcription/regulation effects of chromatin states at the SNP locus[64]. We also conducted Gene Ontology gene-set analyses based on FUMA's gene ontology classification system[29] and pathway

analyses[65] for all mapped genes and the 87 common mapped genes of hippocampal formation. We conducted genome-wide gene-based association and gene-set analyses using MAGMA v.1.08[66] (http://ctg.cncr.nl/software/magma) in FUMA. All variants in the major histocompatibility complex (MHC) region (GRCh37: 6:28,477,797–33,448,354) were excluded before running the MAGMA analyses. MAGMA performs multiple linear regression to map the input SNPs to 18091 protein coding genes and estimates the significance value of that gene. Genes were considered significant if the P value was <0.05 after Bonferroni correction for 18,091 genes. We performed cell type analysis of the 87 genes identified by all four mapping strategies using data available as part of the supplements in Zhang and colleagues 2016[41].

**Multivariate replication analysis**. To ensure that not only single locus associations replicate but that also the multivariate pattern of these associations are consistent in the discovery and replication sample, we implemented a multivariate replication procedure established in Loughnan et al.[67]. In brief, for each locus identified in the multivariate analysis in the discovery sample, this procedure derives a composite score from the mass-univariate $z$-statistics and tests for associations of the composite score with the genotype in the replication sample (for mathematical formulation see Loughnan et al.[67]). Four of the 177 loci could not be tested as the lead SNPs from the discovery sample were not available in the replication sample. For the remaining loci we report the percent of loci replicating at $P < 0.05$ and the percent of loci showing the same effect direction.

**Genetic overlap between hippocampal formation and brain disorders**. We accessed GWAS summary statistics for migraine (MIG) from International headache genetics Consortium[68] and for Parkinson's disease (PD) from the International Parkinson Disease Genomics Consortium[69,70]. The latter included 23andMe data. 23andMe participants provided informed consent and participated in the research online, under a protocol approved by the external AAHRPP-accredited IRB, Ethical & Independent Review Services (E&I Review). Furthermore, from the Psychiatric Genomics Consortium we accessed summary statistics for attention deficit hyperactivity disorder (ADHD)[71], autism spectrum disorder (ASD)[72], bipolar disorder (BIP)[73] and major depression (MD)[74]. Finally, we included data from recent studies of schizophrenia (SCZ)[75] and of Alzheimer's disease (AD)[76]. The included disorders have typical onset times at different phases in life, thus potentially covering a wide range of biological processes affecting the hippocampal formation.

Using conjunctional FDR statistics (FDR < 0.05)[42,43], we identified shared variants associated with hippocampal formation and each of the above-mentioned brain disorders. In contrast to genetic correlation analysis, conjunctional FDR does not require effect directions and can therefore be applied to summary statistics from multivariate GWAS, which do not contain effect directions. Two genomic regions, the extended major histocompatibility complex genes region (hg19 location Chr 6: 25119106–33854733) and chromosome 8p23.1 (hg19 location Chr 8: 7242715–12483982) for all cases and *MAPT* region for PD and *APOE* region for AD and ASD, respectively, were excluded from the FDR-fitting procedures because complex correlations in regions with intricate LD can bias FDR estimation. We submitted the results from conjunctional FDR to FUMA v1.3.7[29] to annotate the genomic loci with conjFDR value < 0.10 having an $r^2 \geq 0.6$ with one of the independent significant SNPs.

**Reporting summary**. Further information on research design is available in the Nature Research Reporting Summary linked to this article.

## Data availability

In this study we used brain imaging and genetics data from the UK Biobank [https://www.ukbiobank.ac.uk/], and GWAS summary statistics obtained from the Psychiatric Genomics Consortium [https://www.med.unc.edu/pgc/shared-methods/], 23andMe [https://www.23andme.com/], International headache genetics Consortium (IHGC) [http://www.headachegenetics.org/content/datasets-and-cohorts], the International Genomics of Alzheimer's Project [https://ctg.cncr.nl/software/summary_statistics], and the International Parkinson Disease Genomics Consortium [https://pdgenetics.org/resources]. The latter included 23andMe data, which was made available through 23andMe under an agreement with 23andMe that protects the privacy of the 23andMe participants [https://research.23andme.com/collaborate/#dataset-access/]. The summary statistics for hippocampal formation derived in this study is available in our github repository [https://github.com/norment/open-science]. FUMA results are available online [https://fuma.ctglab.nl/browse/371].

## Code availability

All code and software needed to generate the results is available as part of public resources, specifically MOSTest (https://github.com/precimed/mostest), FUMA (https://fuma.ctglab.nl/), conjunctional FDR (https://github.com/precimed/pleiofdr) and LD score regression (https://github.com/bulik/ldsc).

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

## Acknowledgements

The authors were funded by the Research Council of Norway (TK: 276082, 323961. OAA: 213837, 223273, 248778, 273291, 262656, 229129, 283798, 311993. LTW: 204966, 249795, 273345), the South-Eastern Norway Regional Health Authority (OAA: 2013-123, 2017-112, 2019-108. LTW: 2014-097, 2015-073, 2016-083), Norwegian Health Association (SB: 22731), Stiftelsen Kristian Gerhard Jebsen, the European Research Council (LTW: ERCStG 802998). The funding bodies had no role in the analysis or interpretation of the data; the preparation, review or approval of the manuscript; nor in the decision to submit the manuscript for publication. This work was performed on the Tjeneste for Sensitive Data (TSD) facilities, owned by the University of Oslo, operated and developed by the TSD service group at the University of Oslo, IT-Department (USIT) and on resources provided by UNINETT Sigma2—the National Infrastructure for High Performance Computing and Data Storage in Norway. The research has been conducted using the UK Biobank Resource (access code 27412) and using summary statistics for various brain disorders that partly included 23andMe data. We would like to thank the research participants and employees of UK Biobank, the 23andMe and the consortia contributing summary statistics for making this work possible.

## Author contributions

S.B. and T.K. conceived the study and analysed the data. K.N. and T.K. interpreted the results and spearheaded the writing. S.B. and T.K. drafted the online methods. A.A.S., O.F., D.v.d.M., A.M.D., L.T.W. and O.A.A. gave conceptual input on the methods and/or results. S.B., K.N., A.A.S., O.F., D.v.d.M., A.M.D., L.T.W., O.A.A. and T.K. contributed to and approved the final manuscript.

## Competing interests

O.A.A. has received speaker's honorarium from Lundbeck and Sunovion, and is a consultant to HealthLytix. A.M.D. is a Founder of and holds equity in CorTechs Labs, Inc., and serves on its Scientific Advisory Board. The terms of this arrangement have been reviewed and approved by UCSD in accordance with its competing interest policies. Other authors report no competing interests.
