## [Peer Review File · Nature Communications]

Distributed genetic architecture across the hippocampal formation implies common neuropathology across brain disordersREVIEWER COMMENTS

Reviewer #1 (Remarks to the Author):

As much of the genetic architecture of the hippocampal formation is currently unknown the authors performed a multivariate genome-wide association analysis in volumetric data from 35,411 individuals, we revealed 173 unique genetic loci with distributed associations across the hippocampal formation. In addition they showed genetic overlap with eight major developmental and degenerative brain disorders, where common genes suggest partly age- and disorder-independent mechanisms underlying hippocampal pathology.

How does the outcome of this study relate to previous genetic analyses of volumetric data of the hippocampus? This does not become clear. The authors only mention the following in the discussion: 'multivariate GWAS of the volumes of the hippocampal formation revealed a 204 plethora of genomic loci not identified in previous work.' and hereafter there is no further elaboration. Moreover I miss mentioning of the study by Horgusluoglu-Moloch et al. Genome-wide association analysis of hippocampal volume identifies enrichment of neurogenesis-related pathways. Sci Rep. 2019 Oct 10;9(1):14498. doi: 10.1038/s41598-019-50507-3.

To assess genetic overlap between hippocampal formation and major brain disorders the authors used conjunctive FDR statistics to identify shared variants associated with hippocampal formation and each of these brain disorders. Why did the authors not assess genetic correlations estimates (r_g) as calculated by linkage disequilibrium score regression between the different disorder which is state-of-the-art?

Downstream analyses of the loci identified and therefore further insight in its biological relevance are quite limited and could be expanded, such as including cell type specific and colocalization analysis. Moreover, as genetic correlations with neurological disorders are shown additional drug targets analyses would also be interesting to perform. The authors already speculate on potential drug targets in their discussion section so such analysis would certainly be a valuable addition.

Moreover it would be interesting to know what the heritability estimates are for the individual hippocampus subregions, these estimated are also missing.

An important part of this study is the analysis of the MRI scans. The authors only write how the MRIs are analyzed with reference to a previous study (TheIglesias, J. E. et al. A computational atlas of the hippocampal formation using ex vivo, ultra-high resolution MRI: Application to adaptive segmentation of in vivo 505 MRI. Neuroimage, 117-137, doi:10.1016/j.neuroimage.2015.04.042 (2015)) . But how reliable is this method? Has it been validated?

The introduction section is relatively long and I advise to shorten it to make it more to the point.

In the online methods section analyses with FUMA are described but in the results also analyses with MAGMA are shown while not described in the methods. Could the authors clarify this.

Reviewer #2 (Remarks to the Author):

In the manuscript "Distributed genetic architecture across the hippocampal formation implies common neuropathology across major brain disorders", Bahrami and colleagues investigate the genetic underpinnings of the hippocampus and its substructures in multivariate regression. The research as such is interesting and important, but in my opinion, the manuscript has several shortcomings which render it difficult to understand.

For instance, page 2, ll. 36-38, ll. 39-42: Overall, the manuscript, could benefit from being revised by a native English speaker. I am myself not a native speaker, but I can clearly identify that there are several problems with the writing that make the manuscript very hard to understand. Please, edit the manuscript thoroughly for language.

Abstract

Page 2, ll. 20-21 and ll. 28-29: You describe "memory formation and decline" and "emotions" as "human behaviours". That's factually wrong. It is important to understand the differences between, behaviour, emotion, and cognition. Please, correct this.

Introduction

You describe the hippocampus and the functions of its subfields in detail in the introduction. I think the manuscript would benefit from a diagram depicting the hippocampus and its subdivisions. Please, add one.

Page 3, l. 47: You describe a "rate of change". What is meant by this? Please, clarify.

Page 3, l. 48: You say "major" psychiatric disorders. What are "minor" psychiatric disorders". I don't think that's accurate. Please, delete.

Page 3, l 51: Please define what you mean by "neurodevelopment". The term is very vague.

Page 3, l 56: It should read "major depressive disorder (MDD)". Please, correct.

Page 3, ll. 59-60: The sentence sounds as if it is incomplete. Please, edit.

Methods

Page 4, l 84: You adjust your hippocampus GWASs for total brain size. With this adjustment you are potentially introducing a collider in your analysis because total brain size lies on the causal path after your outcome. This means that your effect will most probably be from genetic variant -> subfield -> total brain size. If not as main analysis, you must at least perform sensitivity analysis calculating GWASs without adjusting for "intracranial volume".

I find it rather frustrating if I need to read other papers to understand the methods fully. I think that a paper should contain all necessary information to be replicable. Therefore, please, add (at least) a short summary of your MOSTest method.

After reading the MOSTest paper, I was wondering why there is no comparison between the MOSTest method and other multivariate GWAS methods (e.g., MultiPhen)? Why is this not included? I think your paper would benefit from the comparison of your method with other or at least one other multivariate GWAS method to validate your results. Please, add.

You do not calculate or present the SNP-based h^2 of your GWASs. Please add this. As you are having genotypes available, please, use methods based on genotypes (e.g., BOLT-LMM, or GCTA).

You are calculating genetic correlations between the individual subfields of the hippocampus. Why don't you calculate genetic correlations with other previously published GWASs of the hippocampus/brain volume? This would function as an external validation. Please, add this to your manuscript.

Results

You have uploaded your GWAS results to FUMA. However, you are writing that you are making those publicly available when the paper has been accepted for publication. How am I supposed to judge the quality of the GWAS if I cannot access the results on FUMA? Please, make the GWAS public and provide the entry numbers for review.

Page 6, ll. 117-118: What do the chromatin states 1-7 mean? What do the regulatomeDB category 1 or 2 mean. Please, explain and add more detail to the manuscript

Pages 6-7, ll. 124-152; I find the overall presentation of results and discussion tedious. Currently, there is no optimal method to link GWAS hits to actual genes. Several approaches have been chosen by you in your manuscript and overlapping prioritised genes across the methods are discussed in your manuscript. I highly recommend cutting down on these speculative statements. Please, present the results more soberly.

Page 7: You are presenting genetic correlations with other brain disorders. I don't understand reasoning behind the preselection of the disorders. Why not include all PGC brain disorders and other neurological disorder than migraine? Genetic correlation analyses are used to generate new hypothesis, please, include all PGC psychiatric disorders and other neurological disorders which sum stats are published for. Maybe the hippocampus is involved in seizures?

Related to this, it seems that your GWASs did not genetically correlate with any brain disorders which again makes me question the validity of the analytical approach. My guess is, if you do not adjust for intracranial volume, you may detect more of the heritable component and then will be able to detect genetic correlations.

Discussion

I'm missing a limitations section in your manuscript. For example, there has been lots of discussion of the generalisability of the UK Biobank sample to the general population. Please, add a limitations section.

Page 10, l 208: You hypothesise that potential genes common across different phenotypes may be useful drug targets. However, if these genes are shared between several phenotypes may it not also be possible that they have several functions. This would mean manipulating the protein may increase the risk for side effects. Please, discuss this as well.

You are making statements about age dependence and disorder-dependent genetic effects; however, I am not sure if this can be done based on your findings because many of the PGC GWASs contain individuals across the lifespan. Please, reconsider and clarify your statement.

You could take a triangulation approach and use new methods like GenomicSEM or multivariable Mendelian randomisation to investigate disorder-specific effects, please, consider this.

The discussion is a mixture of animal and human studies. I think it is important to make clear which statements are based on animal studies and which on humans. Please, be more specific.

Supplementary Tables

Most of the Supplementary Tables are missing table headers/descriptions and nearly none of the abbreviations are explained. Please, add this.

Christopher Hübel

Reviewer #3 (Remarks to the Author):

In this article, Bahrami and colleagues perform a GWAS of the hippocampal formation with 35,411 individuals from the UK biobank, and report a connection with eight brain disorders. The article is very well written, and is in my opinion a good match for Nat Commun in terms of scope and audience.

I admit that I am far from being an expert in GWAS, but below are some comments on general aspects of the study, as well as on the image analysis methods.

- The narrow ethnicity of the sample (white British) should be mentioned in the abstract, and its effects on the results and limitations should be discussed in the Discussion.
- I miss a thorough discussion of the similarities and differences of the results of this study and those of [22-24].
- Why are the left and right volumes averaged? Wouldn't it be more interesting to potentially discover lateralized relationships?
- Line 84: is the Euler number of the segmentation a well accepted proxy for image quality? (a reference would make **its use more convincing**).

Response to Reviewers

Content	Page #
Response to all reviewers 1
Response to Reviewer #1 2
Response to Reviewer #2 7
Response to Reviewer #3 16

Response to all reviewers

We would like to thank the reviewers for their constructive feedback that clearly helped improving our manuscript. In brief, our major revisions include:

- ✓ We have implemented the latest recommendations for the multivariate GWAS related to quality assurance of the genetics data and software implementations. These changes have improved the genetic signal in the GWAS further (now 177 loci discovered). We have updated this analysis and all downstream analyses of this study, yielding highly similar yet slightly stronger results compared to the previous version.
- ✓ We have added a new replication analysis which implements a recently published procedure specifically tailored for multivariate replication. Using data from 5262 non-white individuals, we significantly replicated our main analysis performed in 35,411 white British individuals.
- ✓ We have added a cell type analysis, which showed that the mapped genes are particularly expressed in astrocytes.
- ✓ We have addressed all individual reviewer comments and have marked all changes in the manuscript related to these comments with **green** colour. Responses to individual reviewer comments are provided on the next pages.

Reviewer #1

Reviewer: As much of the genetic architecture of the hippocampal formation is currently unknown the authors performed a multivariate genome-wide association analysis in volumetric data from 35,411 individuals, we revealed 173 unique genetic loci with distributed associations across the hippocampal formation. In addition they showed genetic overlap with eight major developmental and degenerative brain disorders, where common genes suggest partly age- and disorder-independent mechanisms underlying hippocampal pathology.

Response: We thank the reviewer for the fruitful comments that helped us improve our manuscript.

Reviewer: How does the outcome of this study relate to previous genetic analyses of volumetric data of the hippocampus? This does not become clear. The authors only mention the following in the discussion: ‘multivariate GWAS of the volumes of the hippocampal formation revealed a 204 plethora of genomic loci not identified in previous work.’ and hereafter there is no further elaboration. Moreover I miss mentioning of the study by Horgusluoglu-Moloch et al. Genome-wide association analysis of hippocampal volume identifies enrichment of neurogenesis-related pathways. Sci Rep. 2019 Oct 10;9(1):14498. doi: 10.1038/s41598-019-50507-3.

Response: We thank the reviewer for pointing us at the lack of elaboration on the novelty aspects of this work, and for pointing us to the missing reference. The novelty of this study compared to all previous studies is the multivariate framework which allowed us to discover more of the genetic architecture of the hippocampal formation by capitalizing on the shared genetics of different hippocampus regions. We have now made this clear in the revised version of the manuscript. We would also like to thank the reviewer for pointing us to the study by Horgusluoglu-Moloch et al, which is now included.

In the *Introduction* section, we now state: “*The past decades have brought significant progress towards a characterization of the genetic architecture of the hippocampal formation, from experimental manual mapping in mice¹⁸ to brain imaging based genome-wide-association studies (GWAS) in humans¹⁹⁻²², initially based on total hippocampus volume reporting one²⁰ and six¹⁹ loci, and further increasing to fifteen when studying individual hippocampal subfields²¹. Given the broad functional portfolio of the hippocampal formation, however, it is clear that much of the genetic architecture remains to be explored, calling for further studies and novel analytical approaches²³. Recent work revealed a distributed genetic architecture of human brain anatomy^{24,25} and function²⁶ and suggested that capitalizing on this distributed nature in a multivariate GWAS approach can significantly improve the discovery beyond standard GWAS approaches²⁴. Advancements into deriving subdivisions of the hippocampus through adaptive segmentation has allowed for a fine-grained assessment across multiple subregions²⁷. We hypothesized that the genetic architecture within the hippocampal formation is distributed across its subregions and thus aimed to gain novel insights into the genetics of the*

hippocampal formation by deploying such multivariate GWAS approach to the 19 subregion volumes that can currently be segmented with MRI²⁷.”

In the *Discussion* section we now state: “*Our findings align with and expand upon earlier reports from univariate analyses of the hippocampus^{19,22}, pointing at neurogenesis-related pathways²² and linking hippocampal volumes to common brain disorders^{19,21}.*”

Finally, our *Supplementary Tables* now include information on whether the identified loci are novel or have been previously reported (Suppl. Table 2, Suppl. Table 11-18).

18 Thompson, C. L. et al. Genomic anatomy of the hippocampus. *Neuron* (2008).

19 Hibar, D. P. et al. Novel genetic loci associated with hippocampal volume. *Nat Commun* (2017).

20 Stein, J. L. et al. Identification of common variants associated with human hippocampal and intracranial volumes. *Nat Genet* (2012).

21 van der Meer, D. et al. Brain scans from 21,297 individuals reveal the genetic architecture of hippocampal subfield volumes. *Mol Psychiatry* (2020).

22 Horgusluoglu-Moloch, E. et al. Genome-wide association analysis of hippocampal volume identifies enrichment of neurogenesis-related pathways. *Sci Rep* (2019).

23 Vilor-Tejedor, N. et al. Genetic Influences on Hippocampal Subfields: An Emerging Area of Neuroscience Research. *Neurol Genet* (2021).

24 van der Meer, D. et al. Understanding the genetic determinants of the brain with MOSTest. *Nat Commun* (2020).

25 van der Meer, D. et al. The genetic architecture of human cortical folding. *Sci Adv* (2021).

26 Roelfs, D. et al. Genetic overlap between multivariate measures of human functional brain connectivity and psychiatric disorders. *medRxiv* (2021).

27 Iglesias, J. E. et al. A computational atlas of the hippocampal formation using ex vivo, ultra-high resolution MRI. *Neuroimage* (2015).

Reviewer: To assess genetic overlap between hippocampal formation and major brain disorders the authors used conjunctive FDR statistics to identify shared variants associated with hippocampal formation and each of these brain disorders. Why did the authors not assess genetic correlation estimates (rg) as calculated by linkage disequilibrium score regression between the different disorder which is state-of-the-art?

Response: Genetic correlation analysis requires an effect direction, which is not available in the multivariate framework deployed in this study. Therefore, post-GWAS analyses of multivariate GWAS summary statistics call for statistical approaches that do not require an effect direction, such as for example provided by tools like conjunctive FDR or MAGMA. We have added genetic correlation analysis for the univariate summary statistics of the individual hippocampus regions, where we have effect direction available (Suppl. Figure 6).

It is important to note that the conjunctive FDR framework chosen to study genetic overlap in our main multivariate analysis has a core advantage over genetic correlation analysis. Specifically, it is well possible that a variety of SNPs with opposing effect directions cancel each other out, which would result in low genetic correlation despite significant genetic overlap. Conjunctive FDR in contrast will detect overlap in such scenarios. Indeed, this issue together with the overall weaker genetic signal in the univariate GWAS may explain why Suppl. Figure 6 does not show any strong genetic correlation between disorders and individual hippocampus volumes (none surviving Bonferroni

correction), while conjunctive FDR analysis with the multivariate GWAS of the entire hippocampus formation shows significant overlap with all eight disorders.

We have added the rationale for choosing conjunctive FDR to the *Methods* section of the manuscript: “*In contrast to genetic correlation analysis, conjunctive FDR does not require effect directions and can therefore be applied to summary statistics from multivariate GWAS, which do not contain effect directions.*”

Reviewer: Downstream analyses of the loci identified and therefore further insight in its biological relevance are quite limited and could be expanded, such as including cell type specific and colocalization analysis. Moreover, as genetic correlations with neurological disorders are shown additional drug targets analyses would also be interesting to perform. The authors already speculate on potential drug targets in their discussion section so such analysis would certainly be a valuable addition.

Response: We thank the reviewer for this excellent suggestion. We have now added an analysis of cell types for the genes mapped by all four mapping strategies using data available from Zhang et al (2016). This analysis indicated that the genes were mostly expressed in fetal astrocytes, followed by mature astrocytes and neurons (presented in Suppl. Figure 5, for convenience pasted below). Whereas we agree that an additional drug target analysis would be of high relevance and interest, we prefer to leave this out for a dedicated study focused around drug targets.

Supplementary Figure 5. Cell type analysis for the 87 common genes. Genes that were not expressed or that were missing in the data base were not included. (A) Panel A shows profiles per gene. (B) We scaled profiles within genes and compared them across genes, implicating fetal astrocytes with strongest overall expression. The lower and upper hinges of the boxplot correspond to the first and third quartiles. Pairwise t-test results (** indicates bonferroni significance): fetal astrocytes vs. mature astrocytes $P=5.07e-03$ | fetal astrocytes vs. neurons $P=5.57e-04$ ** | fetal astrocytes vs. oligodendrocytes $P=4.53e-12$ ** | fetal astrocytes vs. microglia/macrophage $P=2.68e-06$ ** | fetal astrocytes vs. endothelial $P=4.49e-10$ ** | mature astrocytes vs. neurons $P=3.01e-01$ | mature astrocytes vs. oligodendrocytes $P=4.44e-08$ ** | mature astrocytes vs. microglia/macrophage $P=1.18e-02$ | mature

astrocytes vs. endothelial $P=6.44e-06$ ** | neurons vs. oligodendrocytes $P=1.57e-04$ ** | neurons vs. microglia/macrophage $P=1.86e-01$ | neurons vs. endothelial $P=1.33e-03$ ** | oligodendrocytes vs. microglia/macrophage $P=1.42e-02$ | oligodendrocytes vs. endothelial $P=9.50e-01$ | microglia/macrophage vs. endothelial $P=4.41e-02$.

41 Zhang, Y. et al. Purification and Characterization of Progenitor and Mature Human Astrocytes Reveals Transcriptional and Functional Differences with Mouse. *Neuron* (2016).

Reviewer: Moreover it would be interesting to know what the heritability estimates are for the individual hippocampus subregions, these estimated are also missing.

Response: We thank the reviewer for pointing this out. We have computed heritability estimates through Genome-wide Complex Trait Analysis (GCTA) and display the results in panel A of Suppl. Figure 1, for convenience pasted below.

Supplementary Figure 1. Heritability of hippocampus volumes and their genetic correlation from univariate GWAS. (A) GCTA-based heritability estimates. The numbers depict h², with standard error in brackets. (B) [...]

Reviewer: An important part of this study is the analysis of the MRI scans. The authors only write how the MRIs are analyzed with reference to a previous study (TheIglesias, J. E. et al. A computational atlas of the hippocampal formation using ex vivo, ultra-high resolution MRI: Application to adaptive segmentation of in vivo 505 MRI. *Neuroimage*, 117-137, doi:10.1016/j.neuroimage.2015.04.042 (2015)). But how reliable is this method? Has it been validated?

Response: The method by Iglesias et al. is implemented as part of the commonly used Freesurfer software tool and has been extensively validated in the referenced paper. Specifically, the authors validated their algorithm in three independent data sets and found robust performance and highly replicable results. We have now made this important point clear in the *Methods* section, where we state “The segmentation method has previously been validated in three independent data sets, attributing robust performance and highly replicable results²⁷.”

27 Iglesias, J. E. et al. A computational atlas of the hippocampal formation using ex vivo, ultra-high resolution MRI: Application to adaptive segmentation of in vivo MRI. *Neuroimage* (2015).

Reviewer: The introduction section is relatively long and I advise to shorten it to make it more to the point.

Response: We have shortened the introduction from initially ~800 words to now ~650 words and have increased clarity where possible. For example, the introductory description of hippocampal anatomy has now been converted into a schematic illustration (Figure 1A) so that we were able to remove the rather long text introducing the individual subregions. We think that the introduction has benefited from the revisions and thank the reviewer for pointing this out.

Reviewer: In the online methods section analyses with FUMA are described but in the results also analyses with MAGMA are shown while not described in the methods. Could the authors clarify this.

Response: We thank the reviewer for pointing us at the lack of information and have added a description of this analysis to the *Methods*: “We conducted genome-wide gene-based association and gene-set analyses using MAGMA v.1.08⁶⁶ (<http://ctg.cncr.nl/software/magma>) in FUMA. All variants in the major histocompatibility complex (MHC) region (GRCh37: 6:28,477,797–33,448,354) were excluded before running the MAGMA analyses. MAGMA performs multiple linear regression to map the input SNPs to 18091 protein coding genes and estimates the significance value of that gene. Genes were considered significant if the *P* value was <0.05 after Bonferroni correction for 18091 genes.”

66 de Leeuw, C. A., Mooij, J. M., Heskes, T. & Posthuma, D. MAGMA: generalized gene-set analysis of GWAS data. *PLoS Comput Biol* (2015).

Reviewer #2:

Reviewer: In the manuscript "Distributed genetic architecture across the hippocampal formation implies common neuropathology across major brain disorders", Bahrami and colleagues investigate the genetic underpinnings of the hippocampus and its substructures in multivariate regression. The research as such is interesting and important, but in my opinion, the manuscript has several shortcomings which render it difficult to understand.

Response: We thank the reviewer for the fruitful comments that helped us improve our manuscript.

Reviewer: For instance, page 2, ll. 36-38, ll. 39-42: Overall, the manuscript, could benefit from being revised by a native English speaker. I am myself not a native speaker, but I can clearly identify that there are several problems with the writing that make the manuscript very hard to understand. Please, edit the manuscript thoroughly for language.

Response: We thank the reviewer for pointing this out. We have carefully revised the manuscript for language and we believe the journal will assist with further language editing – if necessary - during the copy editing of the manuscript.

Reviewer: Abstract. Page 2, ll. 20-21 and ll. 28-29: You describe "memory formation and decline" and "emotions" as "human behaviours". That's factually wrong. It is important to understand the differences between, behaviour, emotion, and cognition. Please, correct this.

Response: We thank the reviewer for pointing out this inapt phrasing. We now write “*Despite its major role in complex human traits across the lifespan, including memory formation and decline, navigation and emotions, much of the genetic architecture of the hippocampal formation is currently unexplored.*”

Reviewer: Introduction. You describe the hippocampus and the functions of its subfields in detail in the introduction. I think the manuscript would benefit from a diagram depicting the hippocampus and its subdivisions. Please, add one.

Response: This is an excellent suggestion. We have integrated the suggested diagram as panel A in Figure 1 and have adjusted panel B such that the colours of the univariate summary statistics of the hippocampus subregions match the respective colouring of the regions in panel A. The new diagram has also allowed us to move the introductory description of hippocampus anatomy to the figure legend of Figure 1, thereby substantially shortening the introduction section (as requested by Reviewer 1). For convenience, we paste the new figure 1 below:

Figure 1. The multivariate framework discovered 177 independent loci significantly associated with the hippocampal formation. (A) Schematic illustration of the hippocampus regions. The hippocampal formation comprises the histologically distinguishable subfields of the hippocampus proper as well as the dentate gyrus with its own subfields, and the neocortical subiculum, presubiculum and parasubiculum. In all but the latter, the hippocampal formation is also divided into an anterior (head) and a posterior part (body). The upper part illustrates the multivariate GWAS statistics for the entire formation with 177 significant loci. The lower part depicts for each of the 177 unique loci the corresponding statistics from univariate GWASs of single subregions (one colour per subregion), supporting a distributed genetic architecture across the hippocampal formation.

Reviewer: Page 3, l. 47: You describe a "rate of change". What is meant by this? Please, clarify.

Response: We have rephrased this to “The degree of hippocampal maturation and change throughout the lifespan [...]”.

Reviewer: Page 3, l. 48: You say "major" psychiatric disorders. What are "minor" psychiatric disorders". I don't think that's accurate. Please, delete.

Response: We have rephrased throughout and now refer to psychiatric disorders (without “major”). Of note, we keep the term “major” for brain disorders (“major brain disorders”), where the term is legitimate to highlight that we are looking at severe and highly prevalent disorders.

Reviewer: Page 3, l. 51: Please define what you mean by "neurodevelopment". The term is very vague.

Response: We have changed this to “development”, which should make clear that we are talking about a time period early in life when typical childhood onset disorders emerge. The sentence now reads “*Indeed, the hippocampal formation is known to be involved in disorders with typical onset during development⁹, such as autism spectrum disorders (ASD), attention-deficit hyperactivity disorder (ADHD), schizophrenia (SCZ) and bipolar disorder (BIP), through its roles in perception, memory processes, modulation of executive function, emotion regulation, among others¹⁰⁻¹²”*

Reviewer: Page 3, l. 56: It should read "major depressive disorder (MDD)". Please, correct.

Response: We here used the term major depression (MD) rather than major depressive disorder (MDD). In the sentence pointed at by the reviewer, the title of the cited work is ‘Course of illness, hippocampal function, and hippocampal volume in major depression’. Moreover, we use the respective manuscript section to introduce the abbreviations for the disorders that we take into genetic analysis. The PGC depression GWAS specifically refers to “major depression (MD)” as they included individuals that have not been clinically diagnosed with the disorder. Therefore, we decided not to implement the suggested change to avoid having to introduce two terms, MDD and MD.

Reviewer: Page 3, ll. 59-60: The sentence sounds as if it is incomplete. Please, edit.

Response: We thank the reviewer for pointing this out and have revised accordingly. The sentence now reads: “*Emerging data suggests a complex hippocampal crosstalk among the dopaminergic and other transmitter systems in PD, where the hippocampal formation is involved in adaptive memory and motivated behaviour¹⁵”*

15 Calabresi, P., Castrioto, A., Di Filippo, M. & Picconi, B. New experimental and clinical links between the hippocampus and the dopaminergic system in Parkinson's disease. *Lancet Neurol* 12, 811-821, doi:10.1016/S1474-4422(13)70118-2 (2013).

Reviewer: Methods. Page 4, l 84: You adjust your hippocampus GWASs for total brain size. With this adjustment you are potentially introducing a collider in your analysis because total brain size lies on the causal path after your outcome. This means that your effect will most probably be from genetic variant -> subfield -> total brain size. If not as main analysis, you must at least perform sensitivity analysis calcuting GWASs without adjusting for "intracranial volume".

Response: Estimated intracranial volume (ICV) is a standard measure used to correct for global differences in head size. ICV is a measure of the cranial cavity and is not a direct measure of total brain size/ brain atrophy. ICV allows us to control for global (brain region unspecific) volumetric differences between individuals. For example, without correcting for intracranial volume, we would find that the average female has hippocampal atrophy in all subfields. Likewise, intracranial volume is correlated to body size so that we would find reduced hippocampus volume in small people on average. Therefore, all volumetric analyses must correct for intracranial volume to avoid systematic bias triggering false conclusions. This is the standard in the field and has been implemented in previous imaging genetics analyses by us and others, including previous univariate GWASs of the hippocampus, e.g. Hibar et al (Nature Communications), Stein et al (Nature Genetics), van der Meer et al (Molecular Psychiatry).

Hibar, D. P. et al. Novel genetic loci associated with hippocampal volume. Nat Commun (2017).

Stein, J. L. et al. Identification of common variants associated with human hippocampal and intracranial volumes. Nat Genet (2012).

van der Meer, D. et al. Brain scans from 21,297 individuals reveal the genetic architecture of hippocampal subfield volumes. Mol Psychiatry (2020).

Reviewer: I find it rather frustrating if I need to read other papers to understand the methods fully. I think that a paper should contain all necessary information to be replicable. Therefore, please, add (at least) a short summary of your MOSTest method. After reading the MOSTest paper, I was wondering why there is no comparison between the MOSTest method and other multivariate GWAS methods (e.g., MultiPhen)? Why is this not included? I think your paper would benefit from the comparison of your method with other or at least one other multivariate GWAS method to validate your results. Please, add.

Response: MOSTest has been extensively validated in the original methods paper, including simulations and comparisons with other methods (MQFAM, MultiPhen, MultiABEL). These validations revealed that MOSTest performed equally well at discovery compared to existing tools, yet with an order of magnitude shorter runtime and better ability to detect cases of invalid type-I error. Since the tool has been extensively validated and has already been used in several recent publications (e.g. van der Meer et al, Science Advances 2021, Shadrin et al, NeuroImage, 2021), we do not see a necessity to validate the method against other tools in this study. Of note, we here include a replication analysis in an independent sample, and the strong replication performance in and of itself serves as a further validation of the MOSTest method.

We thank the reviewer for pointing us at the lack of details introducing the method in our manuscript. We have added the following description to the *Methods* section: “*The resulting residuals for the 20 regions were jointly fed into the multivariate omnibus statistical test (MOSTest)²⁴ analysis.*”

MOSTest implements permutation testing to identify genetic effects across multiple phenotypes, yielding a multivariate GWAS summary statistic across all 20 features. For mathematical details of the implementation, see van der Meer et al (2020)²⁴, for details on the software implementation see github.com/precimed/mostest. MOSTest has been extensively validated in the original methods paper, including simulations and comparisons with other methods that have confirmed its solid performance in discovery and an order of magnitude shorter runtime compared to other tools²⁴.”

Reviewer: You do not calculate or present the SNP-based h2 of your GWASs. Please add this. As you are having genotypes available, please, use methods based on genotypes (e.g., BOLT-LMM, or GCTA).

Response: We thank the reviewer for the suggestion. We have computed heritability estimates through Genome-wide Complex Trait Analysis (GCTA) and display the results in panel A of Suppl. Figure 1, for convenience pasted below.

Supplementary Figure 1. Heritability of hippocampus volumes and their genetic correlation from univariate GWAS. (A) GCTA-based heritability estimates. The numbers depict h2, with standard error in brackets. (B) [...]

Reviewer: You are calculating genetic correlations between the individual subfields of the hippocampus. Why don't you calculate genetic correlations with other previously published GWASs of the hippocampus/brain volume? This would function as an external validation. Please, add this to your manuscript.

Response: We have performed the suggested analysis and have added this sanity check to the legend of Suppl. Figure 1 (where other genetic correlations of univariate summary statistics are reported). We state: “As an additional sanity check we also calculated the genetic correlations between our 20 univariate GWAS summary statistics and the summary statistic of the whole hippocampus from a previous GWAS²¹ and found that all volumes were significantly associated (smallest Rg was with fimbria: Rg=.40, P=6e-07).”

21 van der Meer, D. et al. Brain scans from 21,297 individuals reveal the genetic architecture of hippocampal subfield volumes. *Mol Psychiatry* (2020).

Reviewer: Results. You have uploaded your GWAS results to FUMA. However, you are writing that you are making those publicly available when the paper has been accepted for publication. How am I supposed to judge the quality of the GWAS if I cannot access the results on FUMA? Please, make the GWAS public and provide the entry numbers for review.

Response: We have made our FUMA results public. In the *Data availability* section of the manuscript we now state: “FUMA results are available at <https://fuma.ctglab.nl/browse/371>.”

Reviewer: Page 6, ll. 117-118: What do the chromatin states 1-7 mean? What do the regulomeDB category 1 or 2 mean. Please, explain and add more detail to the manuscript

Response: We thank the reviewer for pointing us at these missing details, which are now described in the legend of *Supplementary Figure 4*: “The chromatin states are 1=Active Transcription Start Site (TSS); 2=Flanking Active TSS; 3=Transcription at gene 5’ and 3’; 4=Strong transcription; 5=Weak Transcription; 6=Genic enhancers; 7=Enhancers; 8=Zinc finger genes & repeats; 9=Heterochromatic; 10=Bivalent/Poised TSS; 11=Flanking Bivalent/Poised TSS/Enh; 12=Bivalent Enhancer; 13=Repressed PolyComb; 14=Weak Repressed PolyComb; 15=Quiescent/Low. RegulomeDB categories reflect: 1a: eQTL + TF binding + matched TF motif + matched DNase Footprint + DNase peak; 1b: eQTL + TF binding + any motif + DNase Footprint + DNase peak; 1c: eQTL + TF binding + matched TF motif + DNase peak; 1d: eQTL + TF binding + any motif + DNase peak; 1e: eQTL + TF binding + matched TF motif; 1f: eQTL + TF binding / DNase peak; 2a: TF binding + matched TF motif + matched DNase Footprint + DNase peak; 2b: TF binding + any motif + DNase Footprint + DNase peak; 2c: TF binding + matched TF motif + DNase peak; 3a: TF binding + any motif + DNase peak; 3b: TF binding + matched TF motif; 4: TF binding + DNase peak; 5: TF binding or DNase peak; 6: Motif hit; 7: Other.”

Reviewer: Pages 6-7, ll. 124-152; I find the overall presentation of results and discussion tedious. Currently, there is no optimal method to link GWAS hits to actual genes. Several approaches have been chosen by you in your manuscript and overlapping prioritised genes across the methods are discussed in your manuscript. I highly recommend cutting down on these speculative statements. Please, present the results more soberly.

Response: We agree with the reviewer that there is currently no optimal method to link GWAS hits to genes, which is why we here cautiously chose to apply four different mapping strategies and discuss only genes that were identified by all four strategies. While this cannot fully overcome all limitations of current gene mapping approaches, we consider it likely that genes identified by all four strategies are robust. To ensure the reader is aware of the limitations, we have now added a statement to the limitation section of the manuscript: “*Some aspects are relevant for interpreting the results of this study. First,*

there is currently no optimal method for gene mapping. We here chose to apply four different mapping strategies and highlighted genes that were identified in all four strategies. While this cannot fully overcome all limitations of current gene mapping approaches, we consider an identification of a given gene by four strategies as an indicator of robustness”

Reviewer: Page 7: You are presenting genetic correlations with other brain disorders. I don't understand reasoning behind the preselection of the disorders. Why not include all PGC brain disorders and other neurological disorder than migraine? Genetic correlation analyses are used to generate new hypothesis, please, include all PGC psychiatric disorders and other neurological disorders which sum stats are published for. Maybe the hippocampus is involved in seizures?

Response: The rationale for choosing the disorders is detailed in the introduction section where we highlight our lifespan approach, using disorders with different typical onset times in life as proxies for potential age-dependent hippocampal pathology. We would like to emphasize that a fair amount of the included disorders are regarded as neurological disorders (Parkinson's disease, Alzheimer's disease, migraine). It is correct that the hippocampus in some cases is involved in seizures, and more specifically in temporal lobe epilepsy. Temporal lobe epilepsy would, however, not fit into the lifespan focus of this study, as it is often caused by brain injury, such as head trauma, stroke, or infections such as encephalitis or meningitis. In the new version of the manuscript, we have carefully revised the introduction to make our rationale to the point.

Reviewer: Related to this, it seems that your GWASs did not genetically correlat with any brain disorders which again makes me question the validity of the analytical approach. My guess is, if you do not adjust for intracranial volume, you may detect more of the heritable component and then will be able to detect genetic correlations.

Response: Lack of genetic correlation does not necessarily indicate lack of polygenic overlap, as shown previously by multiple methods, including LAVA (Werme et al, 2021) and cross-trait MiXeR analysis (Frei et al, 2019). SNPs with opposing effect directions can cancel each other out, which would result in low genetic correlation despite significant genetic overlap. In addition, the weak genetic correlations from univariate analyses provide further evidence that multivariate analysis is an important method to dissect complex interactions. Our multivariate GWAS boosted the signal for discovery. Combined with a statistical tool that can overcome the issue of opposing effect directions cancelling each other out, we here identified significant genetic overlap with all eight disorders.

Frei et al. Bivariate causal mixture model quantifies polygenic overlap between complex traits beyond genetic correlation. Nat Commun (2019)

Werme et al. LAVA: An integrated framework for local genetic correlation analysis. bioRxiv (2021)

Reviewer: Discussion. I'm missing a limitations section in your manuscript. For example, there has been lots of discussion of the generalisability of the UK Biobank sample to the general population. Please, add a limitations section.

Response: We have added a limitation section in which we now discuss generalizability, among other things. *“Third, generalizability of our findings beyond the study population remains to be investigated. We here performed our main analysis in data from 35,411 white British individuals and replicated our findings in data from 5262 individuals with non-white ethnicity, however, both samples were drawn from the same UK Biobank study population.”*

Reviewer: Page 10, l 208: You hypothesise that potential genes common across different phenotypes may be useful drug targets. However, if these genes are shared between several phenotypes may it not also be possible that they have several functions. This would mean manipulating the protein may increase the risk for side effects. Please, discuss this as well.

Response: We thank the reviewer for this comment. Our new limitation section now includes a discussion of the matter: *“Second, we identified genes associated with the hippocampal formation and several of the brain disorders across the lifespan, which might point to shared molecular pathways. However, it has to be emphasized that they could also have different roles in different pathologies. This study lays the foundation for future detailed studies of potential common pathways and drug targets.”*

Reviewer: You are making statements about age dependence and disorder-dependent genetic effects; however, I am not sure if this can be done based on your findings because many of the PGC GWASs contain individuals across the lifespan. Please, reconsider and clarify your statement.

Response: We thank the reviewer for pointing out that the reasoning behind the claim was not clear. Our lifespan approach builds on the fact that different disorders have typical onset times at different phases in life. This is not related to the age of individuals at inclusion into the PGC studies. We have added a clarifying statement to the *Methods section*: *“The included disorders have typical onset times at different phases in life, and we therefore used them as proxies for potential age-dependent hippocampal pathology.”*

Reviewer: You could take a triangulation approach and use new methods like GenomicSEM or multivariable Mendelian randomisation to investigate disorder-specific effects, please, consider this.

Response: We thank the reviewer for the suggestion. We note that such an approach could deliver interesting insights, however, since disorder specificity is a complex topic given lack of sharpness in the diagnostic classification system, overlapping symptoms, comorbidities, among others, we decided to leave such a comprehensive analysis to a dedicated study, to ensure that the complexity can be appropriately addressed.

Reviewer: The discussion is a mixture of animal and human studies. I think it is important to make clear which statements are based on animal studies and which on humans. Please, be more specific.

Response: We thank the reviewer for pointing this out. We have revised the manuscript and now include statements such as "... from experimental manual mapping in mice ...", "... as evident from rodent studies ...", "... Rodent studies also suggest that ..." and "... with animal models suggesting ...".

Reviewer: Supplementary Tables. Most of the Supplementary Tables are missing table headers/descriptions and nearly none of the abbreviations are explained. Please, add this.

Response: We apologize that this has been missing and thank the reviewer for pointing this out. All Supplementary Tables now have headers/legends.

Reviewer #3:

Reviewer: In this article, Bahrami and colleagues perform a GWAS of the hippocampal formation with 35,411 individuals from the UK biobank, and report a connection with eight brain disorders. The article is very well written, and is in my opinion a good match for Nat Commun in terms of scope and audience. I admit that I am far from being an expert in GWAS, but below are some comments on general aspects of the study, as well as on the image analysis methods.

Response: We thank the reviewer for the positive comments that helped us improve our manuscript.

Reviewer: The narrow ethnicity of the sample (white British) should be mentioned in the abstract, and its effects on the results and limitations should be discussed in the Discussion.

Response: We thank the reviewer for pointing this out and have revised the manuscript accordingly:

- In the *abstract* we state: “Here, through multivariate genome-wide association analysis in volumetric data from 35,411 white British individuals, we revealed 177 unique genetic loci with distributed associations across the hippocampal formation.”
- The *results* section describes the replication analysis: “A multivariate replication attempt in an independent sample of 5262 individuals with non-white ethnicity supports robustness of the findings, yielding same effect direction for 98% of the lead SNPs (Supplementary Figure 3).”
- The *Discussion* section covers generalizability aspects: “Third, generalizability of our findings beyond the study population remains to be investigated. We here performed our main analysis in data from 35,411 white British individuals and replicated our findings in data from 5262 individuals with non-white ethnicity, however, both samples were drawn from the same UK Biobank study population.”

Reviewer: I miss a thorough discussion of the similarities and differences of the results of this study and those of [22-24].

Response: We thank the reviewer for pointing us at the lack of elaboration on the novelty aspects of this work. The novelty of this study compared to all previous studies is the multivariate framework which allowed us to discover more of the genetic architecture of the hippocampal formation by capitalizing on the shared genetics of different hippocampus regions. We have now made this clear in the revised version of the manuscript.

In the *Introduction* section, we now state: “The past decades have brought significant progress towards a characterization of the genetic architecture of the hippocampal formation, from experimental manual mapping in mice¹⁸ to brain imaging based genome-wide-association studies (GWAS) in humans¹⁹⁻²², initially based on total hippocampus volume reporting one²⁰ and six¹⁹ loci, and further increasing to fifteen when studying individual hippocampal subfields²¹. Given the broad functional portfolio of the hippocampal formation, however, it is clear that much of the genetic architecture remains to be

explored, calling for further studies and novel analytical approaches²³. Recent work revealed a distributed genetic architecture of human brain anatomy^{24,25} and function²⁶ and suggested that capitalizing on this distributed nature in a multivariate GWAS approach can significantly improve the discovery beyond standard GWAS approaches²⁴. Advancements into deriving subdivisions of the hippocampus through adaptive segmentation has allowed for a fine-grained assessment across multiple subregions²⁷. We hypothesized that the genetic architecture within the hippocampal formation is distributed across its subregions and thus aimed to gain novel insights into the genetics of the hippocampal formation by deploying such multivariate GWAS approach to the 19 subregion volumes that can currently be segmented with MRI²⁷.”

In the *Discussion* section we now state: “*Our findings align with and expand upon earlier reports from univariate analyses of the hippocampus¹⁹⁻²², pointing at neurogenesis-related pathways²² and linking hippocampal volumes to common brain disorders^{19,21}.”*

Finally, our *Supplementary Tables* now include information on whether the identified loci are novel or have been previously reported (Suppl. Table 2, Suppl. Table 11-18).

- 18 Thompson, C. L. et al. Genomic anatomy of the hippocampus. *Neuron* (2008).
- 19 Hibar, D. P. et al. Novel genetic loci associated with hippocampal volume. *Nat Commun* (2017).
- 20 Stein, J. L. et al. Identification of common variants associated with human hippocampal and intracranial volumes. *Nat Genet* (2012).
- 21 van der Meer, D. et al. Brain scans from 21,297 individuals reveal the genetic architecture of hippocampal subfield volumes. *Mol Psychiatry* (2020).
- 22 Horgusluoglu-Moloch, E. et al. Genome-wide association analysis of hippocampal volume identifies enrichment of neurogenesis-related pathways. *Sci Rep* (2019).
- 23 Vilor-Tejedor, N. et al. Genetic Influences on Hippocampal Subfields: An Emerging Area of Neuroscience Research. *Neurol Genet* (2021).
- 24 van der Meer, D. et al. Understanding the genetic determinants of the brain with MOSTest. *Nat Commun* (2020).
- 25 van der Meer, D. et al. The genetic architecture of human cortical folding. *Sci Adv* (2021).
- 26 Roelfs, D. et al. Genetic overlap between multivariate measures of human functional brain connectivity and psychiatric disorders. *medRxiv* (2021).
- 27 Iglesias, J. E. et al. A computational atlas of the hippocampal formation using ex vivo, ultra-high resolution MRI. *Neuroimage* (2015).

Reviewer: Why are the left and right volumes averaged? Wouldn't it be more interesting to potentially discover lateralized relationships?

Response: We here chose to average left and right hemispheres to avoid confounding our multivariate analysis with a set of traits that are very strongly genetically overlapping. This would have yielded a feature set where the relatedness of main interest (i.e. genetic architecture of different regions) would have been confounded with relatedness of hemispheres.

Reviewer: Line 84: is the Euler number of the segmentation a well accepted proxy for image quality? (a reference would make its use more convincing).

Response: We thank the reviewer for pointing out the lack of reference to the Euler number approach. Indeed, work by Rosen and colleagues (*NeuroImage*, 2018) has evaluated quantitative measures of data quality for T1 weighted images and they suggest that Euler number is a robust proxy of image quality.

In one of our own studies (Kaufmann et al, Nature Neuroscience, 2019), we have included a comparison of our own manual quality rating of ~1500 brain images and a Euler-based quality control approach, which confirmed validity of this proxy measure (Suppl. Fig. 13 in that work). We have added reference to the Rosen et al. paper to the current manuscript.

Rosen et al. Quantitative Assessment of Structural Image Quality. Neuroimage (2018).

Kaufmann et al. Common brain disorders are associated with heritable patterns of apparent aging of the brain. Nature Neuroscience. (2019)

REVIEWER COMMENTS

Reviewer #1 (Remarks to the Author):

The authors largely responded to the comments satisfactorily. They only did not follow the advice to perform additional drug target analysis for the following reason: 'Whereas we agree that an additional drug target analysis would be of high relevance and interest, we prefer to leave this out for a dedicated study focused around drug targets.'

Reviewer #2 (Remarks to the Author):

Most of my and other reviewer's points were insufficiently addressed by the authors.

Supplementary Figure 1A. These are unidimensional point estimates. Instead of plotting them as a bar (which implies two dimensions), I recommend plotting them as a point with an error bar for the standard error. Please, change the figure.

Supplementary Figure 1B. It's great that you added the genetic correlations. However, I feel they are not sufficiently discussed. On the one hand, some of these genetic correlations are very high, nearly one; what does this mean for the hippocampal structure? On the other hand, some regions show no correlation at all ($r_g = 0.00$); what is your explanation for this? Do you have a hypothesis? What follow-up investigations would you recommend?

Supplementary Figure 1: None of the abbreviations/acronyms are explained.

Supplementary Figure 6. These genetic correlations do look highly suspicious to me. As the heritability of the different subregions of the hippocampus was estimated with GCTA (based on individual genotypes), it would also be interesting to see the LDSC-estimated h^2 to compare estimates between GCTA and LDSC. Please add, LDSC-based h^2 estimates to supplementary figure 1 for comparison.

You argue that LDSC does not estimate genetic correlations correctly in the presence of local genetic correlations distributed across the whole genome: in the case of supplementary figure 6, this would mean that a nearly equal number of genetic variants would have similar effect sizes in opposite directions to end up with estimates of 0. Looking at your plot, that's an unlikely high probability that exactly this scenario is present so often. Something is off with this analysis, and, please, do investigate what's the reason why you are observing these estimates. Consider creating scatterplots of the effect sizes.

Supplementary Figure 5: None of the abbreviations/acronyms are explained, again.

Describing "memory formation and decline" and "emotions" as human traits is wrong again. Emotions are states and surely not traits. "Memory formation and decline": one is a developmental process and the second a pathophysiological or ageing process depending on the aetiology of the decline. Please correct this factually wrong introduction.

Congratulations on Figure 1. That is a great addition to the manuscript. I would not number the SNPs. The colouring is sufficient. The numbers can be confused with the numbering of the autosomes and do make the figure unnecessarily busy.

There is also no such thing as minor and major brain disorders. It does not make sense. Please, delete. The definition of "major" does not entail "severe and highly prevalent". If you want to emphasise that you are looking at "severe and highly prevalent disorders", then just use exactly use these words in the manuscript.

Instead of using the vague word development, why are you not more specific and use instead: disorders with their typical onset during childhood and adolescence/early life? Humans develop along their whole lifespan (even when they are 50 years old); therefore, it just calling "development" makes no sense.

If the original major depression GWAS does include individuals, that are not clinically diagnosed, then I even recommend avoiding the term "major depression", because it could literally be everything like help-seeking behaviour or other phenotypes. This should be mentioned in your limitations section. Cai, N., Revez, J. A., Adams, M. J., Andlauer, T. F. M., Breen, G., Byrne, E. M., Clarke, T.-K., Forstner, A. J., Grabe, H. J., Hamilton, S. P., Levinson, D. F., Lewis, C. M., Lewis, G., Martin, N. G., Milaneschi, Y., Mors, O., Müller-Myhsok, B., Penninx, B. W. J. H., Perlis, R. H., ... Flint, J. (2020). Minimal phenotyping yields genome-wide association signals of low specificity for major depression. *Nature Genetics*. <https://doi.org/10.1038/s41588-020-0594-5>

Regarding the adjustment of intracranial volume, I do not agree with what the authors have responded. The argument that other studies have been conducted in this way does not automatically mean that alternative routes shouldn't be explored in research. Furthermore, intracranial volume itself is a heritable trait. Maybe, the following paper can convince the authors that they should also conduct a sensitivity analysis and perform GWASs without adjusting for ICV and compare these results with the GWAS adjusted for ICV: Aschard, H., Vilhjálmsdóttir, B. J., Joshi, A. D., Price, A. L., & Kraft, P. (2015). Adjusting for heritable covariates can bias effect estimates in genome-wide association studies. *American Journal of Human Genetics*, 96(2), 329–339. <https://www.sciencedirect.com/science/article/pii/S0002929714005278>

I have checked through the FUMA entry for the GWAS. I'm surprised by the results. Looking at the GTEx results. It looks like the signal of the GWAS is most enriched for cerebellum and pancreas tissue (even though not significant). This makes me wonder if the actual GWAS picks up genetics related to the hippocampus. The authors should comment on these results/discrepancy in their manuscript.

Choosing disorders based on their "average age at onset" is a biased approach and is hindering the progress of science. It is also biased. As I explained in my first response. If you want to investigate age dependence, you will need GWAS summary statistics of GWAS that are based on participants in separate age groups. There is first evidence that different sets of genetic variants may be implicated in disorder onsets at different ages. Therefore, using them as "proxies for potential age-dependent pathology" is not correct.

Our knowledge about the brain and how different substructures of the brain are involved in its pathology is extremely limited. With this genetic correlation analysis, you could contribute by generating new hypothesis discovering associations that are unknown until now. By the way, it is common practice to perform genetic correlation analysis with as many traits as possible. I find it surprising at at some points in your response you argue that you like to follow common practices and here you do exactly the opposite. In the next draft of the manuscript, I expect to see genetic correlations with all psychiatric disorders (btw, anorexia nervosa is also a psychiatric disorder currently missing in your correlation plot) and neurological disorders.

The limitations section should also contain that the GTEx enrichment analysis does not support that your GWAS measures genetic signal of the hippocampus.

I agree with reviewer 3. Please look at potential lateralisation and genetic effects thereof.

Response to Reviewer #2

Reviewer: Supplementary Figure 1A. These are unidimensional point estimates. Instead of plotting them as a bar (which implies two dimensions), I recommend plotting them as a point with an error bar for the standard error. Please, change the figure.

Reviewer: Supplementary Figure 1B. It's great that you added the genetic correlations. However, I feel they are not sufficiently discussed. On the one hand, some of these genetic correlations are very high, nearly one; what does this mean for the hippocampal structure? On the other hand, some regions show no correlation at all ($r_g = 0.00$); what is your explanation for this? Do you have a hypothesis? What follow-up investigations would you recommend?

Reviewer: Supplementary Figure 1: None of the abbreviations/acronyms are explained.

Response: We thank the reviewer for the suggestions on Supplementary Figure 1. We have revised the figure accordingly (for convenience pasted on the next page).

- ✓ We have revised panel A, which now shows point estimates with standard errors as error bars.
- ✓ We have added panel B, a comparison of GCTA-based vs. LDSC-based estimates, to address the second comment by the reviewer (see also our response to that comment below).
- ✓ We have revised the display of statistics in panel C (previously B) to avoid rounding issues (e.g. R_g displayed as 0.0002 rather than 0.00). The fact that some regions are strongly genetically correlated whereas others are not, is not surprising. At the phenotypic level we also observe that some regions are correlated stronger with each other than others. The lowest genetic correlations are with fimbria, which is also the structure with lowest heritability.
- ✓ We have also added explanation of acronyms (h^2 , R_g , GCTA and LDSC) to the legend.

Reviewer: Supplementary Figure 6. These genetic correlations do look highly suspicious to me. As the heritability of the different subregions of the hippocampus was estimated with GCTA (based on individual genotypes), it would also be interesting to see the LDSC-estimated h^2 to compare estimates between GCTA and LDSC. Please add, LDSC-based h^2 estimates to supplementary figure 1 for comparison.

Response: We thank the reviewer for the suggestion and have added LDSC-based estimates to a new panel B in Supplementary Figure 1 (see above). As expected, GCTA-based estimates are generally higher, yet both methods show similar patterns (Pearson correlation between the two: $r=0.97$, $P=4e-13$). We have adjusted the methods description in the *Online Methods* to include the LDSC comparison: “Heritability was estimated using Genome-wide Complex Trait Analysis (GCTA)⁶², and for comparison using LD-score regression^{60,61}.”

Reviewer: You argue that LDSC does not estimate genetic correlations correctly in the presence of local genetic correlations distributed across the whole genome: in the case of supplementary figure 6, this would mean that a nearly equal number of genetic variants would have similar effect sizes in opposite directions to end up with estimates of 0. Looking at your plot, that's an unlikely high probability that exactly this scenario is present so often. Something is off with this analysis, and, please, do investigate what's the reason why you are observing these estimates. Consider creating scatterplots of the effect sizes.

Response: The low genetic correlations between subregions and brain disorders have been reported in previous univariate studies of hippocampus genetics (e.g. Table S5 in van der Meer et al, 2020). As can be observed from Supplementary Fig. 1, the hippocampal subregions are strongly genetically correlated, and therefore our correlation matrix of subregions x disorders cannot be regarded as comprising completely independent associations. This is also reflected in the scatter plot suggested by the reviewer (see below). The left panel figure shows more spread in the dots than the right panel figure, indicating that patterns across brain regions are more similar than patterns across disorders.

Opposing effect directions and their effect on genetic correlation estimates have previously been observed for example in the context of schizophrenia and educational attainment, where studies have consistently shown non-significant correlations yet where genome-wide significant loci have been found to be associated with both phenotypes, and advanced modelling suggests large genetic overlap (see Frei et al., 2019). It is likely that the neurobiological mechanisms underlying these opposing effect directions will apply to a large extent to all brain regions and associated measures.

Van der Meer et al (2020) Molecular Psychiatry. <https://www.nature.com/articles/s41380-018-0262-7>

Frei et al (2019) Nature Communications. <https://www.nature.com/articles/s41467-019-10310-0>

Reviewer: Supplementary Figure 5: None of the abbreviations/acronyms are explained, again.

Response: We thank the reviewer for pointing this out and have added “FPKM= Fragments Per Kilobase Million”.

Reviewer: Describing “memory formation and decline” and “emotions” as human traits is wrong again. Emotions are states and surely not traits. “Memory formation and decline”: one is a developmental process and the second a pathophysiological or ageing process depending on the aetiology of the decline. Please correct this factually wrong introduction.

Response: We thank the reviewer and have now rewritten both sentences to avoid semantic misconceptions. The first sentence of the abstract now reads: “Despite its major role in complex human functions across the lifespan, most notably navigation, learning and memory, much of the genetic architecture of the hippocampal formation is currently unexplored”. The first sentence of the introduction now reads: “The hippocampal formation plays critical roles in episodic memory, navigation, and emotions”.

Reviewer: Congratulations on Figure 1. That is a great addition to the manuscript. I would not number the SNPs. The colouring is sufficient. The numbers can be confused with the numbering of the autosomes and do make the figure unnecessarily busy.

Response: We thank the reviewer for pointing this out. We prefer to keep the numbers in this figure, as it can be difficult to distinguish individual points based on the colours alone. Additional labelling of (a subset of) data points is the recommended practice for scientific data visualization (see e.g. Wilke, Fundamentals of Data Visualisation, <https://clauswilke.com/dataviz/>). Adding the numbers will allow the readers to identify which hippocampal regions show strongest effects. However, we fully agree with the reviewer that potential confusion needs to be avoided and we have added a description of what the numbers reflect: “The strongest effects are labelled with numbers that reflect regions according to the order of regions in the legend (1=CA1_body, 2=CA1_head, ..., 20=Whole_hippocampus).”

Reviewer: There is also no such thing as minor and major brain disorders. It does not make sense. Please, delete. The definition of “major” does not entail “severe and highly prevalent”. If you want to emphasise that you are looking at “severe and highly prevalent disorders”, then just use exactly use these words in the manuscript.

Response: We thank the reviewer for the valuable comment and have revised the phrasing according to the suggestions of the reviewer.

- We have dropped the “major” from the title, abstract and main text.

- The title now reads: “Distributed genetic architecture across the hippocampal formation implies common neuropathology across brain disorders”
- In the introduction, we describe the disorders as “severe and highly prevalent”

Reviewer: Instead of using the vague word development, why are you not more specific and use instead: disorders with their typical onset during childhood and adolescence/early life? Humans develop along their whole lifespan (even when they are 50 years old); therefore, it just calling “development” makes no sense.

Response: We have rephrased to “disorders with typical onset early in life” and “various brain disorders with typical onset at different stages of life”.

Reviewer: If the original major depression GWAS does include individuals, that are not clinically diagnosed, then I even recommend avoiding the term “major depression”, because it could literally be everything like help-seeking behaviour or other phenotypes. This should be mentioned in your limitations section.

Cai, N., Revez, J. A., Adams, M. J., Andlauer, T. F. M., Breen, G., Byrne, E. M., Clarke, T.-K., Forstner, A. J., Grabe, H. J., Hamilton, S. P., Levinson, D. F., Lewis, C. M., Lewis, G., Martin, N. G., Milaneschi, Y., Mors, O., Müller-Myhsok, B., Penninx, B. W. J. H., Perlis, R. H., ... Flint, J. (2020). Minimal phenotyping yields genome-wide association signals of low specificity for major depression. *Nature Genetics*. <https://doi.org/10.1038/s41588-020-0594-5>

Response: Relative to the total size of 135,458 cases in this GWAS, the subset that was included based on self-reported symptoms is relatively small. At least 89% of the full sample cases were clinically diagnosed or self-reported to be in treatment for clinical depression. Furthermore, the authors of this GWAS carefully assessed the comparability of the different included cohorts and report that ‘the common variant genetic architecture of lifetime major depression in these seven cohorts (containing many subjects medically treated for MDD) has strong overlap with that of current depressive symptoms in general community samples’. We prefer to stick with the term ‘major depression’, in line with the labelling in the source GWAS paper, and with the PGC’s phenotype labelling. However, we agree with the reviewer that it is necessary to point the reader at the potential limitation and have therefore added a statement to the limitation section of our manuscript. We now state: “Forth, it should be noted that the here used GWAS for major depression partly included samples that were not necessarily clinically diagnosed with major depressive disorder but reported symptoms of the disorder. This may factor into the specificity of this GWAS.”

Reviewer: Regarding the adjustment of intracranial volume, I do not agree with what the authors have responded. The argument that other studies have been conducted in this way does not automatically mean that alternative routes shouldn't be explored in research.

Furthermore, intracranial volume itself is a heritable trait. Maybe, the following paper can convince the authors that they should also conduct a sensitivity analysis and perform GWASs without adjusting for ICV and compare these results with the GWAS adjusted for ICV:

Aschard, H., Vilhjálmsón, B. J., Joshi, A. D., Price, A. L., & Kraft, P. (2015). Adjusting for heritable covariates can bias effect estimates in genome-wide association studies. *American Journal of Human Genetics*, 96(2), 329–339. <https://www.sciencedirect.com/science/article/pii/S0002929714005278>

Response: Adjusting for ICV is a prerequisite for this analysis that is even denoted in the software package used to derive the hippocampus segmentations, where it says 'hippocampal subfields and amygdala nuclei need to be corrected for ICV'. The correction is performed as part of the imaging analysis. As we have explained in our previous response, without correcting for intracranial volume, we would find that the average female has hippocampal atrophy in all subfields. Likewise, intracranial volume is correlated to body size so that we would find reduced hippocampus volume in small people on average. Of note, many confound factors in brain imaging analysis are heritable. For example, it is inevitable that any analysis of functional brain imaging needs to control for motion confounds, however, motion by itself is a heritable trait (Engelhardt et al., 2017).

Engelhardt et al (2017). *Dev Cogn Neurosci*. <https://pubmed.ncbi.nlm.nih.gov/28223034/>

Reviewer: I have checked through the FUMA entry for the GWAS. I'm surprised by the results. Looking at the GTEx results. It looks like the signal of the GWAS is most enriched for cerebellum and pancreas tissue (even though not significant). This makes me wonder if the actual GWAS picks up genetics related to the hippocampus. The authors should comment on these results/discrepancy in their manuscript.

Response: We thank the reviewer for pointing out this lack of clarity. The GTEx enrichment results available in the FUMA repository are based on all 963 genes mapped from the 177 GWAS loci using either of four different mapping strategies. In our study, we have focused on the 87 genes that were identified by all four strategies to ensure robustness of the results and interpretation. We have detailed this approach in the paper in the section "Functional mapping and annotation identifies 87 genes robustly associated with hippocampal formation". Unfortunately, it is not possible in FUMA to make this analysis available, as FUMA does not offer a functionality to publish *gene2func* results directly. We have therefore previously included the list of genes and all corresponding FUMA results as a table into the supplement of this manuscript. This allows the interested reader to regenerate our FUMA results directly from the list of genes, or to review the patterns from the tables. However, we see now from the reviewer's response that this may not have been clear enough, and have therefore also added the

corresponding figures to the manuscript directly. The new Supplementary Figure (now number 5) is pasted below and shows that the GTEx enrichment analysis based on the 87 genes mapped by all strategies yields significant enrichment for the hippocampus.

Supplementary Figure 5. GTEx enrichment analysis based on the 87 genes mapped by all strategies. (A) General tissue types. (B) Tissue types. The results have been exported from FUMA gene2func.

It is important to note that the signal of the GTEx enrichment analysis depends on several factors such as the background sample size and the number of background genes. This can partly explain why the analysis based on 963 genes that the reviewer mentioned showed enrichment for cerebellum and pancreas tissues. The total sample size for cerebellum is 21% larger and for pancreas it is 46% larger than for the hippocampus, and they include almost twice as many genes. Importantly, our significant finding of the hippocampus for the analysis based on the 87 robustly identified genes even makes a stronger case that we are picking up genes related to the hippocampus because the hippocampus associations are stronger than pancreas and cerebellum (and others) despite smaller sample sizes and smaller number of background genes.

Reviewer: Choosing disorders based on their “average age at onset” is a biased approach and is hindering the progress of science. It is also biased. As I explained in my first response. If you want to investigate age dependence, you will need GWAS summary statistics of GWAS that are based on participants in separate age groups. There is first evidence that different sets of genetic variants may be implicated in disorder onsets at different ages. Therefore, using them as “proxies for potential age-dependent pathology” is not correct.

Response: We thank the reviewer for the opportunity to clarify this. Our aim is not to investigate age dependence per se, but different types of biological processes affecting the hippocampal formation. By

choosing disorders ranging from developmental disorders to neurodegenerative disorders, we make sure to cover a wide range of biological processes affecting the hippocampal formation. We have now removed the phrase “proxies for potential age-dependent pathology”.

Reviewer: Our knowledge about the brain and how different substructures of the brain are involved in its pathology is extremely limited. With this genetic correlation analysis, you could contribute by generating new hypothesis discovering associations that are unknown until now. By the way, it is common practice to perform genetic correlation analysis with as many traits as possible. I find it surprising at at some points in your response you argue that you like to follow common practices and here you do exactly the opposite. In the next draft of the manuscript, I expect to see genetic correlations with all psychiatric disorders (btw, anorexia nervosa is also a psychiatric disorder currently missing in your correlation plot) and neurological disorders.

Response: We now included the proposed exploratory analysis in a separate figure of the supplement, for convenience pasted below. Specifically, we now include genetic correlations with anorexia nervosa (AN), obsessive-compulsive disorder (OCD), posttraumatic stress disorder (PTSD), and generalized epilepsy (GEP). None of the correlations was significant when adjusting for the number of tests.

Supplementary Figure 9. Exploratory analysis, including additional disorders to test for genetic correlation with individual hippocampus regions. Anorexia nervosa (AN⁷⁷), obsessive-compulsive disorder (OCD⁷⁸), posttraumatic stress disorder (PTSD⁷⁹), and generalized epilepsy (GEP⁸⁰). None of the correlations was significant when adjusting for the number of tests.

77 Watson, H. J. et al. Genome-wide association study identifies eight risk loci and implicates metabo-psychiatric origins for anorexia nervosa. *Nat Genet* 51, 1207-1214, doi:10.1038/s41588-019-0439-2 (2019).

78 International Obsessive Compulsive Disorder Foundation Genetics, C. & Studies, O. C. D. C. G. A. Revealing the complex genetic architecture of obsessive-compulsive disorder using meta-analysis. *Mol Psychiatry* 23, 1181-1188, doi:10.1038/mp.2017.154 (2018).

79 Nievergelt, C. M. et al. International meta-analysis of PTSD genome-wide association studies identifies sex- and ancestry-specific genetic risk loci. *Nat Commun* 10, 4558, doi:10.1038/s41467-019-12576-w (2019).

80 International League Against Epilepsy Consortium on Complex, E. Genome-wide mega-analysis identifies 16 loci and highlights diverse biological mechanisms in the common epilepsies. *Nat Commun* 9, 5269, doi:10.1038/s41467-018-07524-z (2018).

Reviewer: The limitations section should also contain that the GTEx enrichment analysis does not support that your GWAS measures genetic signal of the hippocampus.

Response: We have clarified this misunderstanding in our response to one of the previous comments. Our GWAS measures genetic signal of the hippocampus.

Reviewer: I agree with reviewer 3. Please look at potential lateralisation and genetic effects thereof.

Response: As indicated in our response to Reviewer 3, we had methodological reasons for not including both hemispheres individually. By averaging left and right hemispheres we avoided a major confounding factor to our multivariate analysis. Since the genetic architecture of a particular region in a given hemisphere would be strongly overlapping with that region's architecture in the other hemisphere, we would have created a feature set where the relatedness of main interest (i.e. genetic architecture of different regions) would have been confounded with relatedness of hemispheres. An analysis of laterality requires a larger set of analyses and cannot be achieved simply by omitting the averaging step. Such a set of analyses is beyond the volume and scope of this paper. We have now added a sentence to the limitation section that states "Finally, future research targeting lateralization effects may yield additional insight into hemispheric similarities and differences in distributed genetic effects across regions of the hippocampal formation."

REVIEWERS' COMMENTS

Reviewer #2 (Remarks to the Author):

I thank the authors for responding to all my comments thoroughly. I apologise for the delay in my response because I was ill with Covid over the last few weeks. I especially like the additions to supplementary figure one. I also thank the reviewers for clarifying all semantic details. Thank you also for adding the additional supplementary figure 5 which clears up the enrichment results. I think you may have forgotten to reference supplementary figure 9 in-text. I also think the extension of the limitation section is now far clearer. Thank you overall for this fruitful review experience.